



# Multivariate autoregressive modelling and conditional simulation for temporal uncertainty propagation in urban water systems

Jairo Arturo Torres-Matallana[1,2], Ulrich Leopold[2], and Gerard B.M. Heuvelink[1]

[1]Soil Geography and Landscape Group, Wageningen University
[2]Research Group for Sustainable Urban and Built Environment, Department for Environmental Research and Innovation, Luxembourg Institute of Science and Technology

**Correspondence:** Jairo Arturo Torres-Matallana (arturo.torres@list.lu)

**Abstract.** Uncertainty is often ignored in urban water systems modelling. Commercial software used in engineering practice often ignores uncertainties of input variables and their propagation because of a lack of user-friendly implementations. This can have serious consequences, such as the wrong dimensioning of urban drainage systems (UDS) and the inaccurate estimation of pollution released to the environment. This paper introduces an uncertainty analysis framework in urban drainage modelling and applies it to a case study in the Haute-Sûre catchment in Luxembourg. The framework makes use of the EmiStatR model which simulates the volume and substance flows in UDS using simplified representations of the drainage system and processes. A Monte Carlo uncertainty propagation analysis showed that uncertainties in chemical oxygen demand (COD) and ammonium ($NH_4$) loads and concentrations can be large and have a high temporal variability. Further, a stochastic sensitivity analysis that assesses the uncertainty contributions of input variables to the model output response showed that precipitation has the largest contribution to output uncertainty related with water quantity variables, such as volume in the chamber, overflow volume and flow. Regarding the water quality variables, the input variable related to COD in the wastewater has an important contribution to the uncertainty for COD load (66%) and COD concentration (62%). Similarly, the input variable related to $NH_4$ in the wastewater plays an important role in the contribution of total uncertainty for $NH_4$ load (34%) and $NH_4$ concentration (35%). The Monte Carlo simulation procedure used to propagate input uncertainty showed that among the water quantity output variables, the overflow flow is the most uncertain output variable with a coefficient of variation (cv) of 1.59. Among water quality variables, the annual average spill COD concentration and the average spill $NH_4$ concentration were the most uncertain model outputs (coefficients of variation of 0.99 and 0.82, respectively). Also, low standard errors for the coefficient of variation were obtained for all seven outputs. These were never greater than 0.05, which indicates that the selected MC replication size (1,500 simulations) was sufficient. We also evaluated how uncertainty propagation can explain more comprehensively the impact of water quality indicators for the receiving river. While the mean model water quality outputs for COD and $NH_4$ concentrations were slightly above the threshold, the 0.95 quantile was 2.7 times above the mean value for COD concentration, and 2.4 times above the mean value for $NH_4$. This implies that there is a considerable probability that these concentrations in the spilled CSO are substantially larger than the threshold. However, COD and $NH_4$ concentration levels of the river water will likely stay below the water quality threshold, due to rapid dilution after CSO spill enters the river.

**Keywords:** Stochastic sensitivity analysis; uncertainty analysis; input uncertainty; temporal uncertainty; urban water modelling



# 1 Introduction

Combined sewer systems are important components of the urban water infrastructure. These systems are typically found in old and large cities (Baker, 2009; Litrico and Fromion, 2009) and are designed to transport the water generated and accumulated

in an urban catchment to the receiving water body. During normal conditions all water is transported to the treatment facility before it is released to the environment. This is the so-called *throttled outflow* or pass-forward flow (Hager, 2010). However, during extreme conditions with heavy precipitation, the *combined sewer overflow* (CSO) discharges excess water directly to nearby streams, rivers, lakes or other water bodies (Baker, 2009). The CSO contains polluted water and solid matter (Hager, 2010), which, when released to the environment, can have a damaging impact on the water quality status of the receiving waters

(Bachmann-Machnik et al., 2018; Gasperi et al., 2012). CSO pollutant load emissions are of similar or greater magnitude than the emissions from wastewater treatment plants (Gasperi et al., 2012; Bachmann-Machnik et al., 2018). CSO discharge impacts are mainly high peak flows, high organic loads from single events, which can lead to oxygen depletion, and ecotoxic concentrations of ammonia ($NH_3$) (Miskewitz and Uchrin, 2013; Bachmann-Machnik et al., 2018). To reduce pollution in receiving waters it is important to minimise CSO volume.

One of the main variables is chemical oxygen demand (COD), which is an indicator of organic compounds in water. It is used to measure the effluent quality (Viana da Silva et al., 2011). High levels of COD are correlated with a decrease of the amount of dissolved oxygen (DO) available for aquatic organisms. A depletion of DO concentration in the water column from near 9 mg/l (the maximum solubility of oxygen in estuarine water on an average summer day), to below 2 mg/l, is referred to as hypoxia. If hypoxic conditions are reached, the health of the ecosystem is affected, and cause physiological stress, and even

death, to aquatic organisms (on Environmental and atural Resources - CENR, 2003). Ammonium ($NH_4$) is another important variable and is an indicator of nitrogen compounds in water. Concentrations of $NH_4$ in water and wastewater are relevant because high levels of nitrogen in receiving waters can cause eutrophication and, therefore, excessive growth of algae and other micro-organisms, resulting in oxygen dissolved depletion and fish toxicity (Huang et al., 2010).

To better assess environmental impacts, numerical models are applied in urban hydrology to simulate CSO emissions into

the environment. It is recommended, however, that such modelling approaches consider the inherent uncertainty associated with the system representation and the approximation of the model to the reality (Hutton et al., 2011). Moreover, the model inputs are also not free of errors and associated uncertainties will also propagate to the model output (Heuvelink, 1998).

Five approaches to represent uncertainty in the context of urban water systems are often distinguished (Walker et al., 2003; Refsgaard et al., 2007; van Keur et al., 2008; Bach et al., 2014): 1) determinism; 2) statistical uncertainty; 3) scenario uncer-

tainty; 4) recognised ignorance; and 5) total (unrecognised) ignorance. Following van Keur et al. (2008), *determinism* applies when we have knowledge with absolute certainty about the system under analysis. This is the "ideal world" case which is not realistic for urban hydrology systems. The *statistical* approach is useful when it is possible to describe uncertainty in statistical terms, i.e. when uncertainty can be characterised by probability distribution functions (pdfs). The *scenario* approach, in con-





trast, applies when quantitative probabilities cannot be determined, and instead qualitative measures of uncertainty are used.

It is used when possible outcomes of uncertain inputs are known but not the probabilities of these outcomes (Brown, 2004). There is also no claim that the list of possible outcomes (scenarios) is exhaustive. *Recognised ignorance* occurs when there is awareness of lack of knowledge, but without any further possibility to process and address the recognised uncertainty. This is the case of very complex functional or inherently unidentifiable relationships, when e.g. predictions are infeasible due to chaotic behaviour of the system or when our understanding of the system behaviour is too limited (van Keur et al., 2008).

This is common in social systems where behaviour of humans and groups of humans may often unpredictable. Finally, *total ignorance* is the state of *"complete lack of awareness about imperfect knowledge"* (van Keur et al., 2008). It is the opposite of determinism and reflects a state where we do not know that we do not know (Walker et al., 2003). Among the approaches described above, in this paper we will use the statistical approach to characterise and propagate uncertainties.

Three main sources of uncertainty in the context of performance evaluation analysis and design of urban water infrastructure

and urban drainage modelling are identified (Walker et al., 2003; Neumann, 2007; Deletic et al., 2012). First, *model input* uncertainty is related to errors in input data, i.e. in driving forces such as precipitation. Second, *parameter uncertainty* is related to the uncertainty regarding the (calibrated) parameters of the model. Third, *model structural uncertainty* relates to uncertainty due to model conceptualisation and simplification. For instance, an urban drainage model might ignore certain sub-processes such as evaporation or chemical transformation or might simplify a non-linear relation between model variables to a

linear relation. These type of uncertainties are not captured in model input and model parameter uncertainty and are represented by model structural uncertainty. The focus of this work is on the propagation of model input uncertainty.

Regarding methods for uncertainty propagation analysis, a distinction can be made between analytical methods, such as the Taylor series method (Heuvelink, 1998), and numerical techniques, such as Monte Carlo (MC) simulation. Numerical techniques are more flexible and hence more convenient to analyse uncertainty propagation with complex models (Zoppou,

2001). MC simulations are computationally demanding, especially in the case of complex models, but they can still be used if there are sufficient computational resources (Bastin et al., 2013), among others because it can greatly benefit from parallel computing.

Although uncertainty propagation analysis has been applied extensively in hydrologic modelling (e.g. Beven and Binley (1992); Kuczera and Parent (1998); Hutton et al. (2011); Vrugt et al. (2003b, a); Vrugt and Robinson (2007); Renard et al.

(2010); Datta (2011)), the number of applications of long-term simulations in urban drainage modelling is limited and typically does not consider the influence of temporal and spatial correlation in the analysis of propagation of input uncertainty. Temporal correlation occurs in uncertain dynamic variables such as precipitation and COD of household wastewater, because values of these variables over short time lags will be more similar than over large time lags. The same concept applies to variables that are spatially distributed (Webster and Oliver, 2007). It is important to take temporal (and spatial) correlation of uncertain

inputs into account because this may have a major influence on the outcomes of an uncertainty analysis (Heuvelink, 1998). In this paper we perform a temporal uncertainty propagation analysis in urban water modelling, using MC simulation. As a case study we use the simplified model EmiStatR (Torres-Matallana et al., 2018) to predict wastewater volume, COD and NH$_4$





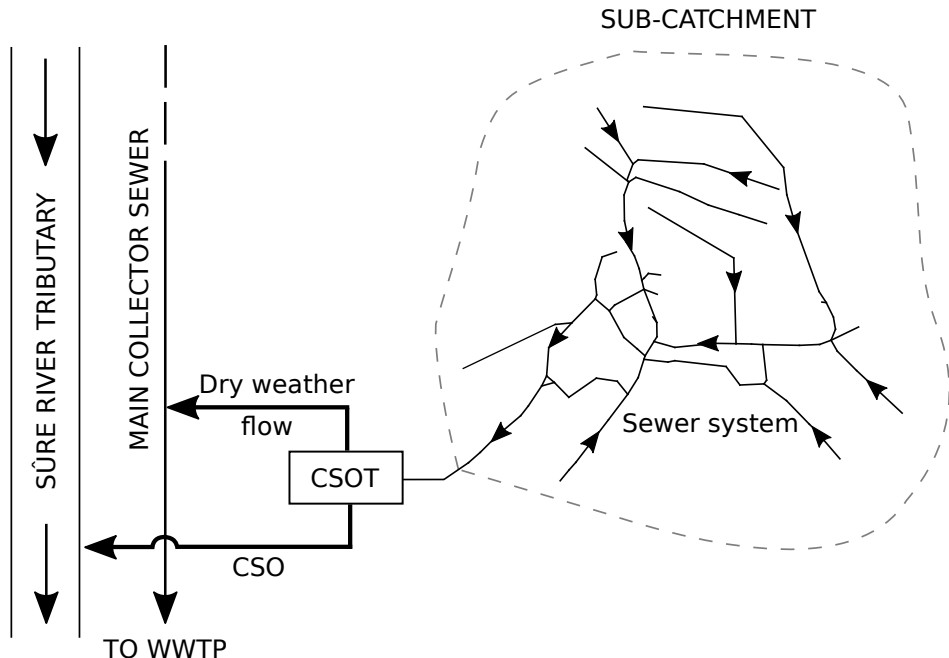

**Figure 1.** Scheme of the sewer system analysed. Adapted from: Andrés-Doménech et al. (2010)

concentrations in CSOs for three urban-rural sub-catchments of the Haute-Sûre catchment in the North-West of the Grand Duchy of Luxembourg.

The objectives of this study are to: 1) select and characterise the main sources of input uncertainty accounting for temporal auto- and cross-correlation within EmiStatR; 2) propagate input uncertainty through EmiStatR, taking into account temporal auto- and cross-correlation of uncertain dynamic inputs; 3) quantify and assess the contributions of each uncertainty source to model output uncertainty dynamically (over time) for the Luxembourg case study.

## 2 Materials and methods

### 2.1 The EmiStatR model

EmiStatR is used to simulate CSO flows and water quality concentrations. Details regarding the conceptual and mathematical model are provided in Torres-Matallana et al. (2018). The main components of the EmiStatR model are: 1) Dry Weather Flow (DWF) including Infiltration Flow (IF); 2) Pollution of DWF; 3) Rain Weather Flow (RWF); 4) Pollution of RWF; 5) Combined Sewer Flow (CSF) and pollution; and 6) Combined Sewer Overflow (CSO) and pollution. Figure 1 illustrates the scheme of the sewer system analysed.

Basically, the total dry weather flow, $Q_{DWF}$ $[\mathrm{l \cdot s^{-1}}]$ is calculated as:





$$Q_{DWF_t} = Q_{s_t} + Q_{f_t} \tag{1}$$

where $Q_{DWF_t}$ $[\mathrm{l}\cdot\mathrm{s}^{-1}]$ is the dry weather flow at time $t$ and $Q_{s_t}$ $[\mathrm{l}\cdot\mathrm{s}^{-1}]$ is the dry weather flow of the residential sewage in the catchment at time $t$, calculated as $86,400^{-1} \cdot pe_t \cdot qs_t$ (where $86,400 = 24 \times 60 \times 60$ is a measurement unit conversion
factor), with $pe_t$ [PE] the population equivalents of the connected CSO structure at time $t$, and $qs_t$ $[\mathrm{l}\cdot\mathrm{PE}^{-1}\cdot\mathrm{d}^{-1}]$ the individual water consumption of households at time $t$. $Q_{f_t}$ $[\mathrm{l}\cdot\mathrm{s}^{-1}]$ is the infiltration flow at time $t$ that enters the pipes from groundwater flow through cracks and joints, calculated as $A_{imp} \cdot q_{f_t}$, where $A_{imp}$ [ha] is the impervious area of the catchment, and $q_{f_t}$ $[\mathrm{l}\cdot\mathrm{s}^{-1}\cdot\mathrm{ha}^{-1}]$ is the infiltration water inflow flux (specific infiltration discharge from groundwater flow) at time $t$. Variables $qs_t$ and $pe_t$ are dynamic and can be defined as time series with daily, weekly and seasonal patterns.

The contribution of rain water to the combined sewage flow, $Q_r$ $[\mathrm{m}^3\cdot\mathrm{s}^{-1}]$, is derived from precipitation as follows:

$$Q_{r_t} = \frac{1}{6} \cdot P_t \cdot [C_{imp} \cdot A_{imp} + C_{per} \cdot (A_{total} - A_{imp})], \tag{2}$$

where $1/6$ is a factor for units conversion, $P_t$ is a time series of precipitation per unit time at time $t$ $[\mathrm{mm} \cdot \mathrm{min}^{-1}]$; $A_{imp}$ is the impervious area of the catchment [ha]; $A_{total}$ is the total area of the catchment [ha]; $C_{imp}$ is the run-off coefficient for impervious areas [-]; and $C_{per}$ is the run-off coefficient for pervious areas [-]. From $Q_{r_t}$, the CSO volume calculation is based
on the exceeding volume stored in the Combined Sewer Overflow Chamber (CSOC). The CSO volume depends on four CSOC stages: (1) filling up; (2) CSO spill volume; (3) stagnation; and (4) emptying. The sum of the total dry weather flow, $Q_{DWF_t}$, and the rain water flow, $Q_{r_t}$, is called combined sewer flow at time $t$, $Q_{CSF_t}$.

The COD load, $B_{COD,Sv}$ [g], in the spill overflow volume is calculated as a function of the spill overflow volume at time $t$, $V_{Sv_t}$ $[\mathrm{m}^3]$, a combined sewer mixing ratio at time $t$, $cs_{mr_t}$ [-], the mean dry weather pollutant concentration at time $t$, $C_{COD_t}$
$[\mathrm{mg}\cdot\mathrm{l}^{-1}]$, and the concentration due to rainwater pollution at time $t$, $COD_{r_t}$ $[\mathrm{mg}\cdot\mathrm{l}^{-1}]$:

$$B_{COD,Sv_t} = (cs_{mr_t} + 1)^{-1} \, V_{Sv_t} (cs_{mr_t} \cdot C_{COD_t} + COD_{r_t}) \tag{3}$$

The variable $V_{Sv_t}$ depends directly on the water volume in the CSO chamber at time $t$, $V_{Chamber_t}$ $[\mathrm{m}^3]$. It is computed as:

$$V_{Sv\,t} = \begin{cases} V_{r_t} + V_{dw_t} - V_{d_t}, & \text{if } V_{Chamber\,t} = V, \\ V_{Chamber\,t} - V & \text{if } V_{Chamber\,t} > V, \\ \epsilon & \text{if } V_{Chamber\,t} < V. \end{cases} \tag{4}$$

where $V_{r_t}$ is the rain weather volume at time $t$ accumulated during a time interval $\Delta t$ [min], $V_{dw_t}$ $[\mathrm{m}^3]$ is the total dry weather
volume (amount of dry weather water in combined sewage flow) at time $t$, $V_{d_t}$ is the volume of throttled outflow to the WWTP at time $t$ $[\mathrm{m}^3]$, $V$ $[\mathrm{m}^3]$ is the CSOC volume, and $\epsilon$ is a numerical precision term set equal to $10^{-5}$ $[\mathrm{m}^3]$. While $V_{Sv}$, $cs_{mr}$ and $C_{COD}$ are dynamic, $COD_{r_t}$ can either be dynamic or assumed constant if the pollution concentration is assumed constant in time. $C_{COD_t}$ $[\mathrm{mg}\cdot\mathrm{l}^{-1}]$ is calculated as (Torres-Matallana et al., 2018):





$$C_{COD\,t} = \frac{10^3 \cdot pe_t \cdot C_{COD,S}}{qs_t \cdot pe_t + 86,400 \cdot A_{imp} \cdot q_{f_t}} \tag{5}$$

where $C_{COD,S}$ is the COD sewage pollution per capita [PE] load per day $[\mathrm{g \cdot PE^{-1} \cdot d^{-1}}]$. Similar equations as above apply to
the second water pollution indicator $NH_4$.

## 2.2  Sewer system in the Haute-Sûre catchment

The study area is composed of three sub-catchments of the Haute-Sûre catchment in the north-west of the Grand-Duchy of
Luxembourg. The combined sewer system drains three villages: Goesdorf (GOE), Kaundorf (KAU), and Nocher-Route (NOR).
The local sewer system downstream each village has a CSO tank to store pollutant peaks in the first flush of combined sewage
flows. Table 2 shows the general characteristics of each CSO tank for each village. Figure 2 depicts the location of the CSO
tanks and the delineation of the sub-catchments. The main land use types in the villages are residential, smaller industries and
farms. Outside of the villages forest as well as agricultural arable and grassland are the dominating land uses. The receiving
water bodies of the CSO structures are tributaries of the river Sûre (Sauer, in German).

## 2.3  Input data

The input variables of the EmiStatR model are shown in Table 1. Following Torres-Matallana et al. (2018), seven input variables
were calibrated: water consumption ($qs_t$), infiltration flow ($q_{f_t}$), flow time structure equivalent to the time of concentration to
the combined sewer overflow tank (CSOT) structure ($t_{fS}$), run-off coefficient for impervious area ($C_{imp}$), run-off coefficient
for pervious area ($C_{per}$), orifice coefficient of discharge ($C_d$), and the initial water level ($Lev_{ini}$). The main objective of the
calibration process is to represent appropriately the water volume in the CSOT.

The observed precipitation ($P_t$) is a one year time series for 2010 at 10 minute time interval, measured at stations Esch-sur-
Sûre and Dahl (Fig. 2). The variable water consumption ($qs_t$) is also dynamic and represented as a time series with a daily
pattern according to factors proposed in the design German guideline ATV-A 134 (Evers et al., 2000).

The hydraulic variable measured is water level in the CSOT $t$, $Lev$ [cm]. The temporal resolution of measurements of $Lev$
is 30 seconds. Regarding wastewater quality (WWQ) characterisation, values of $C_{COD_s}$ and $C_{NH_{4s}}$ in the wastewater were
derived from DWF measurements at Goesdorf, Kaundorf and Nocher-Route. A total of 91 two–hour composite samples were
taken and measured in the laboratory for determination of concentrations of COD $[\mathrm{mg \cdot l^{-1}}]$ and $NH_4$ $[\mathrm{mg \cdot l^{-1}}]$: 7 at Goesdorf
on 4 May 2011, 48 between 19 June and 21 July 2010 at Kaundorf, and 36 between 9 March and 2 August 2011 at Nocher-
Route. The variables $COD_f$ and $NH_{4f}$ were set to zero because the pollution contribution of the infiltration water is negligible
in the study area. The contribution of ammonium from rainwater $NH_{4r}$ was assumed constant and set to 2.00 $[\mathrm{mg \cdot l^{-1}}]$,
while $COD_r$ was equal to zero. Table 1 summarises the base values of the general input variables and Table 2 presents the
base values of input variables for each individual CSO. These base values were used when running EmiStatR in deterministic
mode (see Section 3.1). Some of the variables were calibrated based on observations in the CSOT to simulate water level and
concentrations and loads of pollutants spilled in the CSO to the stream, river or lake.





**Table 1.** General and CSO input, and output variables of EmiStatR. Base values for the general input variables.

| General input | Units | Base value | CSO input | Units |
|---|---|---|---|---|
| *1. Wastewater* | | | *1. Identification* | |
| Water consumption, $qs$ | $[l(PE^a \cdot d)^{-1}]$ | 152 | ID of the structure | [-] |
| Pollution COD[b], $C_{COD,S}$ | $[g(PE \cdot d)^{-1}]$ | 104.2 | Name of the structure | [-] |
| Pollution NH4[c], $C_{NH4,S}$ | $[g(PE \cdot d)^{-1}]$ | 4.7 | *2. Catchment data* | |
| *2. Infiltration water* | | | Name of the municipality | [-] |
| Inflow, $q_f$ | $[l(s \cdot ha)^{-1}]$ | 0.116 | Name of the catchment | [-] |
| Pollution COD, $COD_f$ | $[g(PE \cdot d)^{-1}]$ | 0 | Number of the catchment | [-] |
| Pollution NH4, $NH4_f$ | $[g(PE \cdot d)^{-1}]$ | 0 | Total area, $A_{total}$ | [ha] |
| *3. Rainwater* | | | Impervious area, $A_{imp}$ | [ha] |
| Rain time series, $P$ | [mm] | | Run-off coeff.[d] for impervious area, $C_{imp}$ | [-] |
| Pollution COD, $COD_r$ | $[mg \cdot l^{-1}]$ | 71.0 | Run-off coeff. for pervious area, $C_{per}$ | [-] |
| Pollution NH4, $NH4_r$ | $[mg \cdot l^{-1}]$ | 2.0 | Flow time structure, $t_{fS}$ | [time step] |
| | | | Population equivalents, $pe$ | [PE] |
| | | | *3. CSO structure data* | |
| | | | Volume, $V$ | $[m^3]$ |
| | | | Curve level – volume, $lev2vol$ | [m], $[m^3]$ |
| | | | Initial water level, $Lev_{ini}$ | [m] |
| | | | Maximum throttled outflow, $Q_{d,max}$ | $[l \cdot s^{-1}]$ |
| | | | Orifice diameter, $D_d$ | [m] |
| | | | Orifice coefficient of discharge, $C_d$ | [-] |
| Output variables | | | | |
| *1. Quantity* | | | | |
| Volume in the CSO chamber, $V_{Chamber}$ | $[m^3]$ | | | |
| Overflow spill volume, $V_{Sv}$ | $[m^3]$ | | | |
| Overflow spill flow, $Q_{Sv}$ | $[l \cdot s^{-1}]$ | | | |
| *2. Quality* | | | | |
| Spill COD load, $B_{COD,Sv}$ | [g] | | | |
| Average spill COD conc.[e], $C_{COD,Sv,av}$ | $[mg \cdot l^{-1}]$ | | | |
| 99.9th perc.[f] spill COD conc., $C_{COD,Sv,99.9}$ | $[mg \cdot l^{-1}]$ | | | |
| Maximum overflow COD conc., $C_{COD,Sv,max}$ | $[mg \cdot l^{-1}]$ | | | |
| Spill NH4 load, $B_{NH4,Sv}$ | [g] | | | |
| Average spill NH4 conc., $C_{NH4,Sv,av}$ | $[mg \cdot l^{-1}]$ | | | |
| 99.9th perc. spill NH4 conc., $C_{NH4,Sv,99.9}$ | $[mg \cdot l^{-1}]$ | | | |
| Maximum spill NH4 conc., $C_{NH4,Sv,max}$ | $[mg \cdot l^{-1}]$ | | | |

[a]PE = population equivalents; [b]COD = chemical oxygen demand; [c]NH4 = ammonium;

[d]coef. = coefficient; [d]conc. = concentration; [f]perc. = percentile.





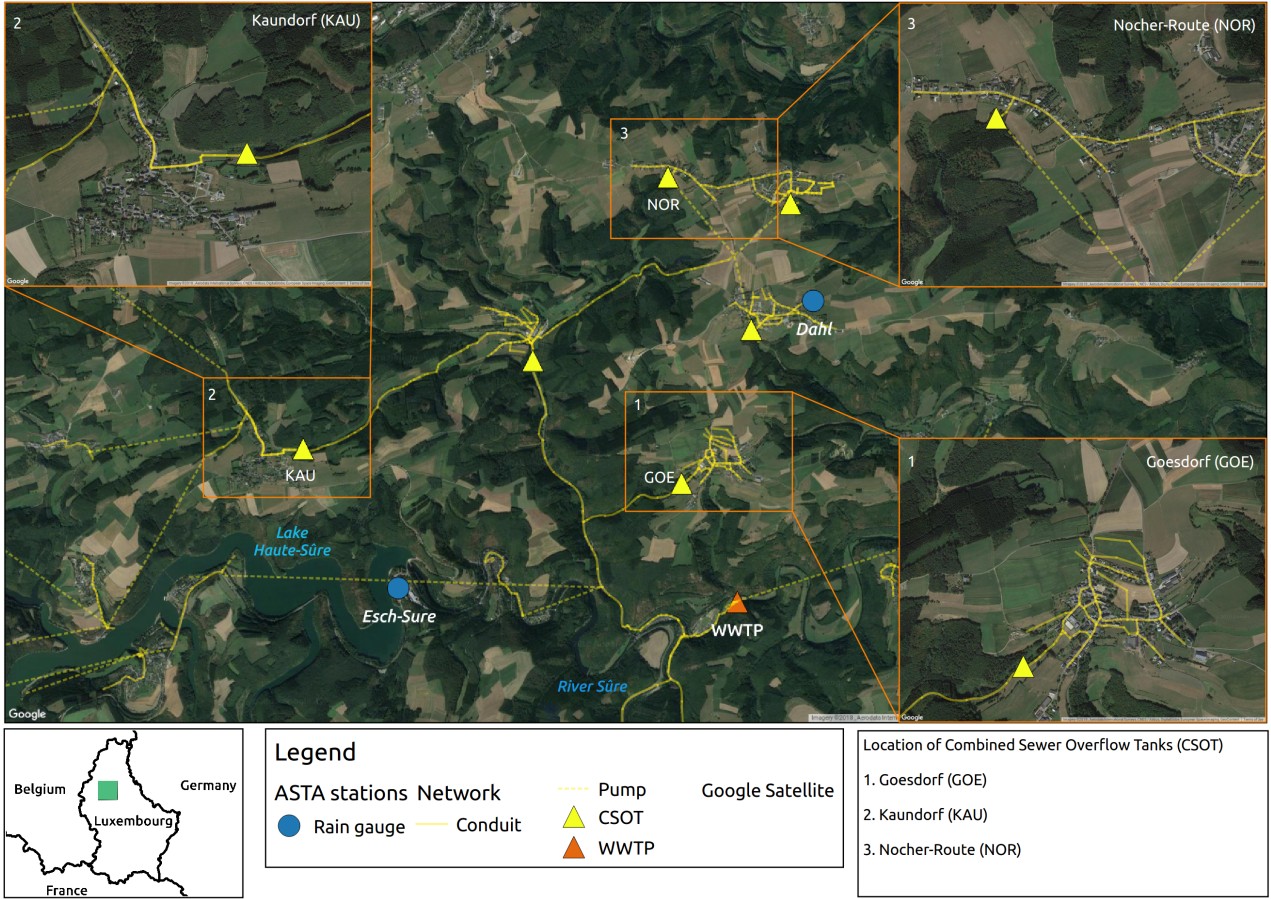

**Figure 2.** The three Haute-Sûre sub-catchments and locations of CSOT structures considered in this study. The background map is provided by © Google Maps.

## 2.4 Selection of model input for uncertainty quantification

Following recommendations from Nol et al. (2010), not all model inputs were taken into account in the uncertainty propagation analysis. Only inputs that are very uncertain and to which the model output is very sensitive were included because these are the ones that have the largest contribution to output uncertainty (Heuvelink (1998), Section 4.4). The level of uncertainty of the inputs was defined by expert judgement and similar case studies in the literature. A quick-scan was used to determine the model sensitivity to each of the model inputs, by running EmistatR in deterministic mode with input base values given in Table 1. The level of model sensitivity was defined by analysing the mathematical model structure and components of the model, expert judgement and simulations with EmiStatR. Inputs that rank high on both the level of uncertainty and on model sensitivity were selected and included in the uncertainty propagation analysis.





**Table 2.** The CSO structure input data for the EmiStatR model, after calibration. For structures 2 and 3, only $C_d$ was calibrated.

| CSO input | | | |
|---|---|---|---|
| *1. Identification* | | | |
| ID of the structure | 1 | 2 | 3 |
| Name of the structure | FBH Goesdorf | FBN Kaundorf | FBH Nocher-Route |
| | | | |
| *2. Catchment data* | | | |
| Name of the municipality | Goesdorf | Kaundorf | Nocher-Route |
| Name of the catchment | Haute-Sûre | Haute-Sûre | Haute-Sûre |
| Number of the catchment | 1 | 1 | 1 |
| Land Use[a] | R/I | R/I | R/I |
| Total area, $A_{total}$ [ha] | 30.0 | 22 | 18.6 |
| Impervious area, $A_{imp}$ [ha] | 5.0 | 11.0 | 4.3 |
| Run-off coefficient for impervious area, $C_{imp}$ [-] | 0.28 | 0.3 | 0.3 |
| Run-off coefficient for pervious area, $C_{per}$ [-] | 0.07 | 0.10 | 0.10 |
| Flow time structure, $t_{fS}$ [min] | 1 | 2 | 2 |
| Population equivalents, $pe$ [PE] | 611 | 358 | 326 |
| | | | |
| *3. CSO structure data* | | | |
| Volume, V [m$^3$] | 190 | 180 | 157 |
| Curve level – volume, $lev2vol$ [m], [m$^3$] | Goesdorf | Kaundorf | Nocher-Route |
| Initial water level, $Lev_{ini}$ [m] | 0.57 | 1.8 | 1.8 |
| Maximum throttled outflow, $Q_{d,max}$ [l · s$^{-1}$] | 5.0 | 9 | 4 |
| Orifice diameter, $D_d$, [m] | 0.15 | 0.20 | – |
| Orifice coefficient of discharge, $C_d$ [-] | 0.67 | 0.67 | 0.67 |

[a] R = residential, I = industrial.

## 2.5 Uncertainty quantification of selected model input

Because we used a statistical approach, probability distribution functions (pdfs) are the basis to represent uncertainties of the selected model inputs. This constitutes the most difficult step of an uncertainty analysis and is done in different ways for constants and dynamic variables, as explained in the following sub-sections.

### 2.5.1 Uncertain constants

Following Heuvelink et al. (2007), an uncertain continuous numerical constant $C$ can be characterised by its marginal (cumu-
lative) pdf (mpdf):

$$F_C(c) = P(C \leq c) \tag{6}$$

Usually a parametric approach can be taken, meaning that a common shape for $F_C$ is chosen (e.g., normal, lognormal, exponential, uniform) so that the mpdf is reduced to a number of parameters. In this study, the input variables that are in this category





are: water consumption ($qs$), infiltration inflow ($q_f$), total area ($A_{total}$), impervious area ($A_{imp}$), the run-off coefficients for

impervious area ($C_{imp}$) and pervious area ($C_{per}$), population equivalents ($pe$), flow time structure ($t_{fS}$), and initial water level ($Lev_{ini}$).

### 2.5.2   Univariate autoregressive modelling

Dynamic uncertain inputs may be temporally autocorrelated. This may dramatically influence the outcome of an uncertainty propagation analysis and must therefore be accounted for. One way of doing this is by assuming an autoregressive order one

(AR(1)) model:

$$y_t = \mu + \phi\,(y_{t-1} - \mu) + w_t, \qquad t = 1,2,...,T, \quad y_0 \sim \mathcal{N}(\mu,\sigma^2) \tag{7}$$

where $y_t$ is the uncertain input at time $t$, $\mu$ is its mean, $\phi$ is the autoregressive parameter ($0 \le \phi < 1$), and $w_t$ is a Gaussian white noise time series with mean zero and variance $\sigma_w^2$. The initial value $y_0$ is taken from a normal distribution with mean $\mu$ and variance $\sigma^2$. The parameters of the model can be estimated based on observations, or in absence of observations, suitable

values are taken based on expert judgment or literature reference values. Note that the effect of the initial condition usually fades out quickly and hence is not of important concern.

    The implementation of the AR(1) model in R was done via the R function arima.sim of the R base package stats (R-Core-Team and contributors worldwide, 2017), both for model calibration and simulation.

### 2.5.3   Multivariate autoregressive modelling

In case of multiple uncertain dynamic inputs, cross-correlation between these inputs may also need to be included. For example, $C_{COD,S}$ and $C_{NH4,S}$ and their uncertainties are likely correlated. This can be done using a multivariate AR(1) model (Luetkepohl, 2005), which is a natural extension of the univariate AR(1) model:

$$Y(t+1) = \boldsymbol{\mu} + \mathbf{A} \cdot [Y(t) - \boldsymbol{\mu}] + \varepsilon(t), \qquad t = 1,2,...,T, \quad Y_0 \sim \mathcal{N}(\boldsymbol{\mu}, \boldsymbol{\Lambda}) \tag{8}$$

where $Y(t)$ is a vector of inputs at time $t$, $\mathbf{A}$ is a square matrix with parameters that define how the variables at time $t+1$

depend on those at time $t$, $\boldsymbol{\mu}$ is now a vector of means and $\varepsilon(t)$ a vector of zero-mean, normally distributed white noise processes. We further assume that the variance-covariance matrix $\mathbf{C}$ of $\varepsilon(t)$ is time-invariant. The initial value $Y_0$ is assumed normally distributed and uncorrelated ($\boldsymbol{\Lambda}$ is a diagonal matrix). In order to estimate the vector $\boldsymbol{\mu}$ and matrices $\mathbf{A}$ and $\mathbf{C}$, a sample of the variables of interest is needed. Parameter estimation is done by means of the R-package mAr (Barbosa, 2015).

### 2.5.4   Input precipitation model

In case precipitation is selected as an uncertain input to be included in the uncertainty analysis, then it too must be characterised by a pdf. Since precipitation, however, is not normally distributed and has many zeros, we cannot make the Gaussian assumption and hence we cannot use the approach described in Section 2.5.2 to model its dynamic behaviour and uncertainty. In addition,





we usually have precipitation measurements nearby so we need to condition the simulations to these measurements. Recall from Section 2.3 that in the case study precipitation data are recorded at stations Esch-sur-Sûre and Dahl.

Torres-Matallana et al. (2017) present a model to simulate precipitation inside a target catchment given a known precipitation time series in a nearby location outside the catchment, while accounting for the uncertainty that is introduced due to spatial variation in precipitation. The method used for input precipitation uncertainty characterisation is essentially the same as the application of a Kalman filter/smoother (Kalman, 1960; Webster and Heuvelink, 2006). Calibration of that model requires precipitation time series at two locations near the catchment of interest. We briefly summarise the method here. We denote the

measured time series of precipitation at the first location as $P_1(t)$ and that at the second location as $P_2(t)$. Once the model is calibrated, it is used to simulate precipitation inside the target catchment from a single precipitation time series nearby the catchment.

*Calibration*

We begin by relating the two precipitation time series as:

$P_1(t) = P_2(t) \cdot \delta(t)$                                                          (9)

where $\delta(t)$ is a positive multiplicative factor that varies over time. We assume that $P_1(t)$, $P_2(t)$ and $\delta(t)$ are stationary and log-normally distributed stochastic processes. After log-transformation we get

$$\log(P_1(t)) = \log(P_2(t)) + \log(\delta(t)) \tag{10}$$

We apply a Kernel (daniell) smoothing to the precipitation time series to avoid rapid fluctuation of the time series for

precipitation depth values smaller than 0.1 mm. This also solves problems associated with taking logarithms of near-zero values. Next, in order to estimate the parameters of $\delta(t)$, we filter the time series allowing the computation of a ratio between the two measured time series. This ratio represents the difference in precipitation as registered in two nearby rain gauge stations. It is computed only for those cases where the precipitation depth of the two time series is greater than 0.01 mm.

     To simplify notation we write $LP_1(t) = \log(P_1(t))$, $LP_2(t) = \log(P_2(t))$ and $L\delta(t) = \log(\delta(t))$. Since two out of three

determine the third, we need only define two processes. We model the joint distribution of $LP_1(t)$ and $L\delta(t)$ by a bivariate AR(1) process, as introduced before:

$$\begin{bmatrix} LP_1(t+1) \\ \\ L\delta(t+1) \end{bmatrix} = \begin{bmatrix} \mu_1 \\ \\ \mu_\delta \end{bmatrix} + \begin{bmatrix} B_{11} & B_{12} \\ \\ B_{21} & B_{22} \end{bmatrix} \left( \begin{bmatrix} LP_1(t) \\ \\ L\delta(t) \end{bmatrix} - \begin{bmatrix} \mu_1 \\ \\ \mu_\delta \end{bmatrix} \right) + \begin{bmatrix} \varepsilon_1(t+1) \\ \\ \varepsilon_\delta(t+1) \end{bmatrix} \tag{11}$$

where $\varepsilon_1$ and $\varepsilon_\delta$ are zero-mean, cross-correlated and normally distributed white noise processes.





To calibrate this model, i.e. estimate its parameters $\mu_1$, $\mu_\delta$, $B_{11}$, $B_{12}$, $B_{21}$, $B_{22}$, $\sigma_1^2$, $\sigma_\delta^2$ and $\rho_{1\delta}$, where $\sigma_1^2 = \text{var}(\varepsilon_1)$,
$\sigma_\delta^2 = \text{var}(\varepsilon_\delta)$ and $\rho_{1\delta}$ is the correlation between $\varepsilon_1$ and $\varepsilon_\delta$, we used the R package mAr (Barbosa, 2015). Calibration is based
on two time series of $LP_1$ and $L\delta$ derived from observed time series $P_1$ and $P_2$.

*Conditional simulation*

To simulate a time series $P$ for the target catchment from an observed time series $P_o$ at a nearby location, we make use of the
fact that the calibrated AR(1) model quantifies how precipitation at one location relates to that at a nearby location. We make
use of Eq. 9:

$$P(t) = P_o(t) \cdot \delta(t) \tag{12}$$

This requires simulations of $\delta(t)$. These are obtained using the calibrated model Eq. 11, but now applied to the vector
$[LP_o \; L\delta]^T$, which characterises the joint pdf of $LP_o$ and $L\delta$. We use this model to simulate $L\delta$ conditional to the observed
time series $LP_o$. Since the two processes are jointly normally distributed we can make use of a well known property of the
multivariate normal distribution (Searle, 1997, page 47). Let $U$ and $V$ be two jointly normally distributed random vectors. The
conditional distribution of $U$ given $V = v$ is then also normal and given by:

$$\{U|V = v\} \sim N\left(E[U] + cov(U,V) \cdot var(V)^{-1} \cdot (v - E[V]), \; var(U) - cov(U,V) \cdot var(V)^{-1} \cdot cov(V,U)\right) \tag{13}$$

We make use of this equation to simulate $\delta$ by substituting:

$$U = L\delta(t+1) \qquad\qquad V = \begin{bmatrix} L\delta(t) \\ LP_o(t+1) \\ LP_o(t) \end{bmatrix} \tag{14}$$

for all $t = 1, ..., T$, while substituting the observed time series $LP_o$ for $v$. For details we refer to Torres-Matallana et al. (2017).

## 2.6 Uncertainty propagation

Various methods can be used to analyse uncertainty propagation. de Rocquigny et al. (2008) summarised the characteristics
of the main methods, which range from deterministic methods such as minimum/maximum to hydrid methods as First- and
Second-Order Reliability Methods (FORM/SORM), considering as well the Taylor series approximation and Monte Carlo
(MC) simulation. We used MC simulation (Hammersley and Handscomb, 1964; Kalos and Whitlock, 2008) to analyse how
input uncertainty propagates through the EmiStatR model, because it is flexible and straightforward to implement. It is also
feasible in our case study because EmistatR is a relatively simple model that does not involve a long computation time.





### 2.6.1 Monte Carlo simulation

The MC method runs the EmiStatR model repeatedly, each time using different model input values, sampled from their pdf.
The method thus consists of the following steps:

1. Repeat $n$ times:

    (a) Generate a set of realisations of the uncertain model inputs at 10 min resolution

    (b) For this set of realisations, run the model at 10 min resolution and store output. Later, in order to compute the
    summary statistics, a temporal aggregation of the model output to one hour intervals is done.

2. Compute and store sample statistics from the $n$ model outputs.

Here, $n$ is the number of MC runs, i.e. the MC sample size. Common sample statistics that measure the uncertainty are the
standard deviation and quantiles of the distribution of MC outputs, such as the difference between the 0.95 and % 0.05 quantile,
which can be easily calculated from the $n$ Monte Carlo outputs.

Sampling from the pdf of uncertain inputs was done using simple random sampling.

### 2.6.2 Monte Carlo output summary

Proper presentation of MC outputs is important to get the most out of the experiment. Therefore, summary statistics are one
important way to summarise the MC outputs. Commonly, a MC study yields $n$ model outputs, which are stored in the MC result
matrix $\mathbf{X}$ in Boos and Osborne (2015). From this matrix, various statistics can be computed. Basic summary statistics include
the mean $\mu_{\mathrm{MC}}$, the standard deviation ($\sigma_{\mathrm{MC}}$) and the variance $\sigma^2_{\mathrm{MC}}$. From these we can compute the coefficient of variation
$CV_{\mathrm{MC}}$ ($\sigma_{\mathrm{MC}}/\mu_{\mathrm{MC}}$), which is a dimensionless expression of relative uncertainty. The coefficient of variation is a standardised
measure of the spread of a sampling distribution, being useful because it allows to directly compare variation in samples with
different units, or with very different means (Marwick and Krishnamoorthy, 2019). We computed estimates and standard errors
for these statistics and also for the interquartile range (IQR$_{\mathrm{MC}}$), 0.005 ($\zeta_{0.005}$) and 0.995 ($\zeta_{0.995}$) quantiles, and the 99% width
of the prediction band ($\zeta_{w,0.99}$).

### 2.6.3 Bootstrap computation for Monte Carlo summary

Following Boos and Osborne (2015) "Good statistical practice dictates that summaries in MC studies should always be ac-
companied by standard errors", we used the bootstrap method to compute standard errors of all MC statistics. These tools are
particularly relevant in a case without analytic solutions (Boos, 2003). According to Boos and Osborne (2015, p. 228) standard
errors for MC output statistics are often not computed, being an additional computational step on top of the overall analysis.
Standard errors are straightforward to compute for simple statistics such as the sample mean over the replications of the MC
output, but are more difficult to compute for more complex statistics, such as medians, sample variances and the classical
Pearson measures of skewness and kurtosis. Therefore, to avoid burdensome computations we opted to compute the standard



errors by the bootstrap method. We briefly explain the bootstrap method below. For a detailed explanation we refer to Efron (1979).

To compute the bootstrap variance of estimators we follow the logic given by Boos and Osborne (2015). From a MC sample $Y_1, ..., Y_n$, we draw a random sample of size $n$ with replacement $Y_1^*, ..., Y_n^*$, and compute an estimator $\hat{\theta}$ of the MC statistic $\theta$ from this resample. We independently repeat this process $B$ times, resulting in a sample of estimators $\hat{\theta}_1, ..., \hat{\theta}_B$. Then the bootstrap variance estimate, $\hat{V}_B$, is the sample variance of this sample of estimators:

$$\hat{V}_B = \frac{1}{B-1} \sum_{i=1}^{B} \left( \hat{\theta}_i - \bar{\hat{\theta}} \right)^2 \tag{15}$$

where $\bar{\hat{\theta}}$ is the mean of the sample of estimators. The MC standard error, se, is simply the square root of the bootstrap variance.

We implemented in stUPscales (Torres-Matallana et al., 2019) specific routines for computing, by means of the bootstrap method, the MC estimators and their standard error for all MC statistics, where the variance of the model output is the most important. We compared our results with the results obtained using the Monte.Carlo.se R-package (Boos et al., 2019).

### 2.6.4   Contributions of input variables to total uncertainty

A number of $m + 1$ MC analysis are needed to compute the contributions of input variables to total uncertainty, where $m$ is the number of model input variables selected for uncertainty quantification. The first MC analysis, $MC_{tot}$, is done to compute the total output uncertainty by varying stochastically all input variables. The uncertainty associated with the first variable $x_1$ is quantified by a second MC analysis $MC_1$, in which only $x_1$ is equal to its deterministic value, while the other input variables vary stochastically. Similarly, the other MC simulations $MC_2$, $MC_3$, ..., $MC_m$ are used to quantify the uncertainty for the
variables $x_2$, $x_3$, ..., $x_m$.

To quantify the contributions of individual input variables to the total uncertainty of the model inputs, the stochastic sensitivity $S_i$ for each uncertain input $x_i$ is computed. The first-order stochastic sensitivity index $S_i$ is defined as (Saltelli et al., 2008, p. 160-161):

$$S_i = \frac{\mathrm{Var}(MC_i)}{\mathrm{Var}(MC_{tot})} \tag{16}$$

The first-order stochastic sensitivity index represents the main effect contribution of each input factor to the total variance of the output. The larger the index, the more important the input uncertainty. We computed stochastic sensitivity indices per time step and aggregated contributions for the whole year. For plotting purpose, we aggregated the outputs from 10 minutes time step to hourly time steps. The aggregation was done for each individual MC run before the contributions were computed.

### 2.7   Water quality impact

The results of the MC uncertainty propagation were also compared with the water quality standards. Standards are introduced to evaluate the impact of emissions of COD and $NH_4$ in CSOs into the receiving water. However, as Toffol (2006) recognises,





although there are European emission standards for wastewater treatment plant effluent, standards for combined sewer over-flow are not so clear. According to Steinel and Margane (2011), the European Water Framework Directive (WFD) is mainly concerned with the natural state of waters. Therefore, emission standards for effluent discharge are not set. The EU Directive

91/271/EEC (1991) sets standards for COD and total Nitrogen, hence similar values have been adopted in many European member states. For more details about guidelines and design procedures in Europe see Blumensaat et al. (2012). We assessed the emissions accordingly to the German guideline ATV-A 128 (1992), which is the standard for dimensioning and design of stormwater structures in combined sewers and commonly used in Luxembourg. The Austrian ÖWAV-RB 19 (2007) is also taken into account because it provides key reference guidelines for design of urban water infrastructure in central Europe.

Three main indicators are taken into account: hydraulic impact, COD concentration, and acute ammonium toxicity.

### 2.7.1 Hydraulic impact

According to the Austrian guidelines and as summarised by Kleidorfer and Rauch (2011), the evaluation of the hydraulic impact is given by:

$$Q_1 \leq f_h \cdot Q_{r1} \tag{17}$$

where $0.1 \leq f_h \leq 0.5$, $Q_1$ [l·s$^{-1}$] is the maximum sewer overflow discharge with return period one year, and $Q_{r1}$ [l·s$^{-1}$] is the maximum water discharge in the river with return period once per year. The factor $f_h$ is taken as 0.1 in more sensitive streams, whereas it is 0.5 for streams with more stable bed and higher re-colonisation potential Toffol (2006). Time series of daily values recorded in 2006 to 2013 of the river Sûre at Heiderscheidergrund were used to compute the daily flow expected with return period once per year (1.01 years), $Q_{r1}$.

According to the German guideline ATV-A 128 (1992), the throttle discharge at CSOs, $Q_{t,CSO}$ [l·s$^{-1}$], may be computed using:

$$Q_{t,CSO} = f_t \cdot A_{imp} \tag{18}$$

where $7.5 \leq f_t \leq 15$ and $A_{imp}$ [ha] is the impervious area connected to the combined sewer system. For the overflow flow MC output mean, 0.95 quantile and 0.995 quantile, we computed the exceedance percentage over the thresholds, calculated as the

proportion of time steps exceeding the number of total time steps in the year (8,759 time steps at 1-hour).

### 2.7.2 COD concentration

Steinel and Margane (2011, Table 14) presents the effluent standards for discharging into freshwater adopted in selected European countries. A COD concentration of 125 [mg · l$^{-1}$] is reported for the European Union countries. Austria has stricter rules with a standard of 90 [mg · l$^{-1}$] for populations between 50 and 500 inhabitants, and 75 [mg · l$^{-1}$] for populations greater than

500 inhabitants. The Goesdorf population by 2001 was 1,025 inhabitants and by 2011 was 1,297 inhabitants (Statec, 2020). For NH$_4$ a similar approach was used.





### 2.7.3   Acute ammonium toxicity

Following Kleidorfer and Rauch (2011), "*the ammonia (NH$_3$) concentration depends on the ammonium (NH$_4$) concentration and on the dissociation equilibrium between NH$_3$ and NH$_4$ (which is influenced by temperature and pH-value)*". According

to Kleidorfer and Rauch (2011), the Austrian guideline ÖWAV-RB 19 (2007) establishes a maximum value of 2.5 mg $\cdot$ l$^{-1}$ for the ammonium (NH$_4$) concentration calculated for one hour duration for salmonid streams. For cyprinid streams a maximum value of 5.0 mg $\cdot$ l$^{-1}$ is recommended.

## 3   Results

### 3.1   Selection of model inputs for uncertainty quantification

In this section we assess the degree of uncertainty and sensitivity for all input variables, following the procedure described in Section 2.4. We summarise the results in Tables 3 and 4.

### 3.1.1   Wastewater

Water consumption, $qs$, is a fairly uncertain input variable and the model output is sensitive to this variable. Volume and flow of CSO are sensitive to changes in $qs$. Regarding water quality output, total load of NH$_4$ is very sensitive to changes in $qs$.

Pollution of sewage as COD load per capita per day, $C_{COD,S}$ is the first selected input variable for propagation of uncertainty, due to the fact that it is both a very uncertain input variable and the model output (average and 99.9 percentile overflow COD concentration) is very sensitive to it. Pollution of sewage as NH$_4$ load per capita per day, $C_{NH4,S}$, is also included in the uncertainty propagation analysis. It is a very uncertain input variable and the model output (overflow load and concentrations of NH$_4$) is very sensitive to it. The variables $C_{COD,S}$ and $C_{NH4,S}$ are very uncertain because these are correlated to the

temporal and spatial pattern of water consumption, which has a daily, weekly and seasonal temporal variability.

### 3.1.2   Infiltration water

Inflow of infiltration water, $q_f$ is a very uncertain input variable because this inflow depends of the number of anomalies in the pipes (cracks or wrong connections) that allow infiltrations flowing into and out of the system. The distribution of these anomalies has a strong random component and hence $q_f$ is very uncertain, and model output is sensitive to it. Pollution of

infiltration water as COD load per capita per day, $COD_f$ and pollution of infiltration water as NH$_4$ load per capita per day, $NH4_f$ are not uncertain because in the Haute-Sûre study area the values of these variables are negligibly small.

### 3.1.3   Rainwater

Precipitation, $P$ is the main driving force of the model and given the spatial variability of the rain fields, this input is considered very uncertain. The model output, additionally, is very sensitive to it. As a consequence, this input variable is treated as the

third input variable in the uncertainty propagation analysis. Pollution of runoff as COD concentration, $COD_r$ is the fourth





input variable considered in the uncertainty propagation, given that it is a very uncertain and very sensitive input variable, particularly to load and concentration of COD in the overflow. Pollution of runoff as NH$_4$ concentration, $NH4_r$ is considered fairly uncertain. The model output (overflow load and concentration of NH$_4$) is very sensitive to it.

### 3.1.4 Sub-catchment

The model is very sensitive to the total area $A_{total}$ and to the run-off coefficient for pervious area ($C_{per}$) and sensitive to the impervious area $A_{imp}$ and to the run-off coefficient for impervious area ($C_{imp}$). However, we did not include $A_{total}$ and $C_{per}$ in the uncertainty analysis because these can be fairly accurately derived from spatial databases and hence their uncertainty is not large. The population equivalents $pe$ is a sensitive variable but not very uncertain. Hence this variable was not included in the uncertainty analysis. The theoretical largest flow time in the catchment $t_{fS}$ is not uncertain and not sensitive.

### 3.1.5 CSO structure

Although model output is very sensitive to maximum throttled outflow $Q_{d,max}$ and volume $V$, these are not included in the uncertainty analysis because their values are accurately known. The same is true for the variables curve level - volume $lev2vol$, orifice diameter $D_d$ and discharge coefficient $C_d$. These variables are accurately known and therefore not considered as uncertain variables. The initial water level in the chamber $Lev_{ini}$ is very uncertain but the model output is not sensitive to
this variable. Therefore, $Lev_{ini}$ was not included in the uncertainty analysis.

### 3.2 Uncertainty quantification of selected model input

After evaluation of the model output sensitivity and taking into account the degree of uncertainties of each input, we selected four input variables to be included in the uncertainty analysis. These are $C_{COD,S}$, $C_{NH4,S}$, $COD_r$ and $P$ (Table 4).

### 3.2.1 Sewage per capita COD and Ammonium

The fit of pdfs for the two uncertain inputs $C_{COD,S}$ and $C_{NH4,S}$ was based on measurements under dry weather flow conditions. Measurement campaigns were done in Goesdorf from 28th April to 24th June 2011, in Kaundorf from 22nd June to 18th August in 2010 and from 20th July to 5th August in 2011, and in Nocher-Route from 18th November 2010 to 27th April 2011. Samples of $COD$ and $NH_4$ in mg·l$^{-1}$ (91 in total for each variable) were analysed. An average wastewater amount was calculated for Goesdorf (153 l·PE$^{-1}$·d$^{-1}$), Kaundorf (112 l·PE$^{-1}$·d$^{-1}$) and Nocher-Route (94.3 l·PE$^{-1}$·d$^{-1}$). Table 5 presents
summary statistics of the dry weather flow measurements of COD and NH4 and the corresponding value of $C_{COD,S}$ and $C_{NH4,S}$. COD is converted to $C_{COD,S}$ by means of a simple conversion from mg·l$^{-1}$ to g·PE$^{-1}$·d$^{-1}$, by multiplying COD by the measured per capita flow (112 l·PE$^{-1}$·d$^{-1}$) and dividing by 1,000. NH$_4$ was converted to $C_{NH4,S}$ in a similar way.

Closer inspection showed that $C_{COD,S}$ and $C_{NH4,S}$ observations are best characterised by a lognormal distribution (Fig. 3). Since $C_{COD,S}$ and $C_{NH4,S}$ are dynamic and cross-correlated, we calibrated a bivariate AR(1) model with state vector $Y =$
$[\log(C_{COD,S}) \quad \log(C_{NH4,S})]^T$. The estimated parameters of the model using the methodology described in Section 2.5.3 are:




**Table 3.** Results of deterministic sensitivity analysis. Average percentage of change of model output caused by $\pm 10\%$ change in model inputs ($qs$, $C_{COD,S}$, $C_{NH4,S}$, $COD_r$, $pe$, and $P$ as time series, VAR(1) model for $C_{COD,S}$ and $C_{NH4,S}$, AR(1) model for $COD_r$ and AR(1) conditioned for $P$. See Table 1 for nomenclature definition). Output change greater than 15% is considered very high. Variable $C_d$ (not shown in the table) leads to a percentage of change less than 0.3%, while variables $t_{fS}$ and $Lev_{ini}$ (not shown in the table) lead to no change in the output. Values greater than 15 are shown in bold font.

| Output variable | $qs$ | $C_{COD,S}$ | $C_{NH4,S}$ | $q_f$ | $COD_r$ | $NH4_r$ | $A_{total}$ | $A_{imp}$ | $C_{imp}$ | $C_{per}$ | $pe$ | $Q_{d,max}$ | $V$ | $D_d$ | $P$ |
|---|---|---|---|---|---|---|---|---|---|---|---|---|---|---|---|
| $V_{Chamber}$ | 4.2 | 0.0 | 0.0 | 3.0 | 0.0 | 0.0 | 8.6 | 7.1 | 5.9 | 7.2 | 4.2 | **16.7** | 7.5 | 0.8 | 13.4 |
| $V_{Sv}$ | 2.9 | 0.0 | 0.0 | 1.7 | 0.0 | 0.0 | **19.6** | 11.6 | 13.5 | **16.1** | 2.9 | 13.1 | **16.5** | 0.2 | **17.8** |
| $Q_{Sv}$ | 0.6 | 0.0 | 0.0 | 0.2 | 0.0 | 0.0 | 2.7 | 1.5 | 1.1 | 1.8 | 0.6 | 13.1 | 14.8 | 0.6 | 12.4 |
| | | | | | | | | | | | | | | | |
| $B_{COD,Sv}$ | 2.9 | 2.4 | 0.0 | 1.7 | 7.7 | 0.0 | **20.1** | 11.8 | 13.7 | **16.6** | 5.3 | **15.7** | 14.5 | 0.2 | **20.7** |
| $C_{COD,Sv,Av}$ | 0.7 | 4.6 | 0.0 | 0.5 | 5.4 | 0.0 | 0.5 | 0.6 | 0.6 | 0.7 | 4.0 | 3.2 | 4.0 | 0.2 | 1.1 |
| $C_{COD,Sv,99.9}$ | 1.6 | 6.7 | 0.0 | 0.8 | 3.4 | 0.0 | 2.8 | 2.3 | 1.9 | 2.4 | 5.1 | 0.0 | 0.0 | 0.0 | 9.2 |
| $C_{COD,Sv,Max}$ | 1.6 | 6.7 | 0.0 | 0.8 | 3.4 | 0.0 | 2.8 | 2.3 | 1.9 | 2.4 | 5.1 | 0.0 | 0.0 | 0.0 | 9.2 |
| | | | | | | | | | | | | | | | |
| $B_{NH4,Sv}$ | 3.1 | 0.0 | 3.3 | 1.8 | 0.0 | 6.7 | **20.4** | 12.0 | 13.8 | **16.8** | 6.4 | **17.0** | 13.4 | 0.3 | **22.1** |
| $C_{NH4,Sv,Av}$ | 0.9 | 0.0 | 5.8 | 0.6 | 0.0 | 4.2 | 0.6 | 0.8 | 0.9 | 0.9 | 5.3 | 4.3 | 5.4 | 0.2 | 1.5 |
| $C_{NH4,Sv,99.9}$ | 1.6 | 0.0 | 7.6 | 0.8 | 0.0 | 2.4 | 3.5 | 2.6 | 2.3 | 2.9 | 6.1 | 0.0 | 0.0 | 0.0 | 11.3 |
| $C_{NH4,Sv,Max}$ | 1.6 | 0.0 | 7.6 | 0.8 | 0.0 | 2.4 | 3.5 | 2.6 | 2.3 | 2.9 | 6.1 | 0.0 | 0.0 | 0.0 | 11.3 |

$$\boldsymbol{\mu} = \begin{bmatrix} 4.40947 \\ 3.70411 \end{bmatrix} \qquad \mathbf{A} = \begin{bmatrix} 0.99165 & -0.00319 \\ -0.00009 & 0.99455 \end{bmatrix} \qquad \mathbf{C} = \begin{bmatrix} 0.00913 & 0.00224 \\ 0.00224 & 0.00185 \end{bmatrix} \qquad (19)$$

The defined multivariate autoregressive model also capture the dynamic behaviour, temporal correlation and cross-correlation of the input variables, deriving the probability distributions of $C_{COD,S}$ and $C_{NH4,S}$ from measurements in the Haute-Sûre catchment, which agreed well with values reported in the literature (Katukiza et al., 2014; Heip et al., 1997).

### 415 3.2.2 Runoff COD concentration

Regarding $COD_r$, due to the fact that no field measurements were available, expert judgement and reference values from the literature were the basis to characterise the pdf of this input variable. The variable was assumed to be lognormally distributed with a mean value of 71 [mg·l$^{-1}$]. Although, House et al. (1993) and Welker (2008) reported a higher value, 107 [mg·l$^{-1}$] for $COD_r$, we selected a lower value due to the specific characteristics of the CSO system in the Haute-Sûre catchment. The value 420 of 150 [mg·l$^{-1}$] as standard deviation of $COD_r$ leads to a coefficient of variation (sd·mean$^{-1}$) equal to 2.11, which is greater than the coefficient of variation for $C_{COD,S}$ (0.84). We allow the standard deviation of $COD_r$ to be greater than the standard deviation of $C_{COD,S}$, because COD measurements in rain water are very uncertain.





**Table 4.** Input variables of the EmiStatR model and selection of inputs for uncertainty analysis based on input uncertainty level and model sensitivity level (legend: from ++ very uncertain/sensitive to – – not uncertain/sensitive).

| Input variable | Input uncertainty | Model sensitivity | Uncertainty analysis |
|---|:---:|:---:|:---:|
| *Wastewater* | | | |
| 1. $qs$ | + | + | no |
| 2. $C_{COD,S}$ | ++ | ++ | **yes** |
| 3. $C_{NH4,S}$ | ++ | ++ | **yes** |
| | | | |
| *Infiltration water* | | | |
| 4. $q_f$ | ++ | + | no |
| 5. $COD_f$ | – – | – – | no |
| 6. $NH4_f$ | – – | – – | no |
| | | | |
| *Rainwater* | | | |
| 7. $P$ | ++ | ++ | **yes** |
| 8. $COD_r$ | ++ | ++ | **yes** |
| 9. $NH4_r$ | + | ++ | no |
| | | | |
| *Sub-catchment* | | | |
| 10. $A_{total}$ | + | ++ | no |
| 11. $A_{imp}$ | + | + | no |
| 12. $C_{imp}$ | + | + | no |
| 13. $C_{per}$ | + | ++ | no |
| 14. $pe$ | + | + | no |
| 15. $t_{fS}$ | – | – – | no |
| | | | |
| *CSO structure* | | | |
| 16. $Q_{d,max}$ | – | ++ | no |
| 17. $V$ | – | ++ | no |
| 18. $D_d$ | – – | – – | no |
| 19. $C_d$ | – – | – – | no |
| 20. $Lev_{ini}$ | ++ | – – | no |





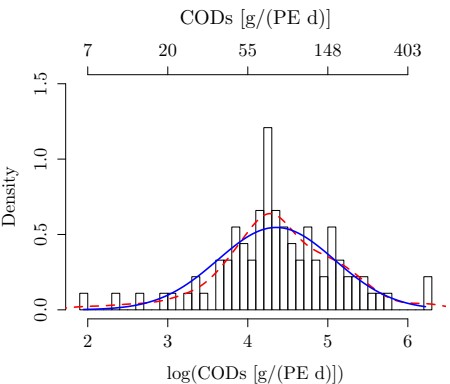

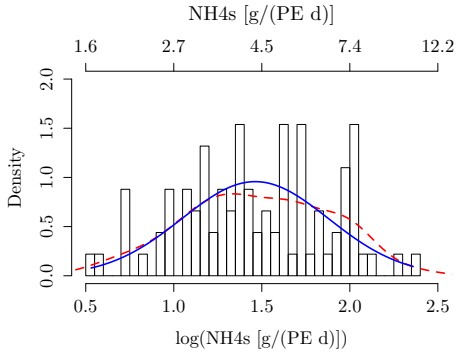

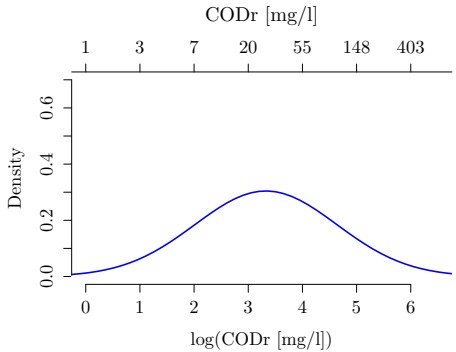

**Figure 3.** Histogram of observations, empirical density (red dashed line) and theoretical normal density (blue line) for (a) $log(C_{COD,S})$; (b) $log(C_{NH4,S})$; (c) $log(COD_r)$.





**Table 5.** Summary statistics of dry weather flow measurements for $C_{COD,S}$ and $C_{NH4,S}$ characterisation.

|  | COD | $C_{COD,S}$ | $\log(C_{COD,S})$ | NH4 | $C_{NH4,S}$ | $\log(C_{NH4,S})$ |
|---|---|---|---|---|---|---|
|  | [mg·l$^{-1}$] | [g·PE$^{-1}$·d$^{-1}$] | log(g·PE$^{-1}$·d$^{-1}$) | [mg·l$^{-1}$] | [g·PE$^{-1}$·d$^{-1}$] | log(g·PE$^{-1}$·d$^{-1}$) |
| Min | 61.9 | 6.9 | 1.936 | 16.10 | 1.745 | 0.556 |
| P5 | 216.8 | 23.8 | 3.167 | 20.55 | 2.102 | 3.018 |
| Mean | 925.5 | 104.2 | 4.378 | 44.38 | 4.733 | 1.473 |
| P95 | 2032.0 | 236.8 | 5.466 | 79.00 | 7.684 | 2.039 |
| Max | 3454.0 | 528.5 | 6.270 | 81.20 | 10.771 | 2.377 |
| St. deviation | 631.7 | 87.5 | 0.751 | 18.56 | 1.917 | 0.410 |

### 3.2.3   Input precipitation model

Precipitation and its associated uncertainty was modelled as an autoregressive model conditioned to the observed precipitation
at a nearby measurement station. We assumed a multivariate lognormal distribution and included temporal correlation of the
simulated time series. Calibration of the precipitation model is done with the mAr package as explained in Section 2.5.4 and
using 10-minute precipitation time series of stations Esch-sur-Sûre and Dahl for 2010. Upon calibration of the multivariate
autoregressive model, we proceeded with the conditional simulation of $Y_c$ (Eq. (13)). For this, we computed the parameters of
the model as shown in Eq. (20). The model parameters are given by (Torres-Matallana et al., 2017):

$$
\quad
\begin{aligned}
\mu_1 &= 2.85501 \\
\mu_\delta &= 0.10194
\end{aligned}
\qquad
\mathbf{B} =
\begin{bmatrix}
0.95650 & 0.03980 \\
0.02429 & 0.88304
\end{bmatrix}
\qquad
\begin{aligned}
\sigma_1^2 &= 0.07241 \\
\sigma_\delta^2 &= 0.07951 \\
\rho_{1\delta} &= -0.03876
\end{aligned}
\qquad (20)
$$

Next we generated conditional simulations of the 10-minute precipitation for 2010 for each subcatchment using the approach
described in Section 2.5.4. Note that this involves simulating log-transformed precipitation which can easily be transformed
to precipitation data using the antilog. The simulation procedure was repeated as many times as simulated precipitation time
series were required for the MC uncertainty propagation analysis.

The simulated precipitation time series captured the main statistics of the observed time series well. Despite the satisfactory
performance of the proposed method, some cases showed an overestimation of the simulated precipitation, mainly due to high
values of the ratio of the multiplicative factor $\delta(t)$. This behaviour was also recognised by McMillan et al. (2011), who stated
that the multiplicative factor used in their study *"does not capture the distribution tails, especially during heavy precipitation
where input errors would have important consequences for runoff prediction"*.

### 440   3.3   Uncertainty propagation

Model output sensitivity and the degree of uncertainties evaluation of each model input helped to define the four input variables
included in the uncertainty analysis: $C_{COD,S}$, $C_{NH4,S}$, $COD_r$ and $P$. In this section we present the results of the uncertainty





propagation for these four selected input variables to the model output, both for water quantity (volume in the combined sewer overflow tank, CSOT, and overflow volume and flow) and for water quality (loads and concentrations of chemical oxygen demand, COD, and ammonium, NH$_4$).

### 3.3.1 Monte Carlo simulation size

In order to perform the MC propagation analysis, we first did a convergence test to estimate the number of simulations required. Besides this test, we also computed the standard error of all MC outputs. These two methods have the same aim and are closely related. In the convergence test, the standard deviation of two different MC simulations with different random seeds were computed and compared for the seven output variables of EmiStatR, three representing water quantity variables ($V_{Chamber}$, $V_{Sv}$ and $Q_{Sv}$) and four for water quality ($B_{COD,Sv}$, $B_{NH4,Sv}$, $C_{COD,Sv,av}$, and $C_{NH4,Sv,av}$). The results of the test indicated that in most cases between 250 and 1,000 MC simulations are enough to reach stable results in terms of the Nash–Sutcliffe model efficiency coefficient (NSE), where a NSE of 1 means a perfect match between observations and model output. In this case we got a NSE $\approx 0.998$ for overflow volume. Regarding the water quality variable $B_{COD,Sv}$, the test showed that a larger number of MC simulations is required. Between 1,000 and 2,000 simulations are required to reach stable results (NSE $\approx 0.880$ for overflow COD load and 0.998 for overflow NH$_4$ load). Therefore, a number of 1,500 MC simulations was used to perform the uncertainty analysis of the water quantity and water quality outputs. Figure 4 illustrates results of the convergence test for the cases where the number of MC replications is 250, 1,000 and 1,500. In this figure the MC1 output is plotted on the x-axis and MC2 output on the y-axis. Although the model output corresponds to yearly time series at 10 minutes resolution, we only plotted those points where the overflow magnitude, and therefore COD and NH$_4$ load, is different from zero. As an indication, for a MC replication size of 1,500, the NSE values for overflow COD and NH$_4$ concentrations are 0.816 and 0.998, respectively.

The computing times per MC replication are presented in Table 6. The computations were performed with two different Linux machines, a laptop with four cores for simulations between 50 and 500 replications, and a server with 80 cores for performing the simulations above 500 replications. Similar execution times were reached for MC1 and MC2 for one-month time series at 10-min time steps (August 2010, 4,464 time steps), while substantial differences were obtained when the 80 cores server was used. We obtained similar timing for 1,500 replications with 50 cores as for 250 replications using three cores in the laptop. The timing reached demonstrates the feasibility to perform a solid MC uncertainty propagation analysis with EmiStatR.

### 3.3.2 Monte Carlo output and uncertainty quantification

The seven output variables from EmiStatR were analysed by MC input uncertainty propagation. Figure 5 illustrates the uncertainty propagation outcomes for the first Monte Carlo simulation, where all input variables vary stochastically. The MC simulations were performed for the entire year 2010 at 10 minutes time step, which were aggregated to hourly time steps in the figure. The aggregation function used for precipitation, CSO chamber volume, CSO spill volume and loads was the sum, whereas for CSO spill flow and concentrations the aggregation function it was the mean. The figure (top) shows input precipi-



**Figure 4.** Results of the MC convergence test for (a, b, c) volume in overflow; (d, e, f) overflow COD load; (g, h, i) overflow NH$_4$ load. Each open circle refers to a ten minute time instant in 2010. As an indication, for a MC replication size of 1,500, the NSE values for overflow COD and NH$_4$ concentrations are 0.816 and 0.998, respectively. Dotted line is the 1:1 line. SD = Standard Deviation.





**Table 6.** Average running time in minutes for Monte Carlo (MC) replications and specific cores used with two different seeds for the pseudo-random number generator in R. The rainfall input used was a one year length time series with 10 minutes time steps from 1 to 31 August 2010 (4,464 time steps).

| Replications | 250 | 500 | 1000 | 1500 | 2000 |
|---|---|---|---|---|---|
| cores | 3 | 3 | 50 | 50 | 50 |
| MC1 | 7.12 | 14.23 | 4.84 | 7.33 | 9.40 |
| MC2 | 7.09 | 14.63 | 4.96 | 7.26 | 9.53 |
| Average | 7.10 | 14.43 | 4.90 | 7.29 | 9.46 |

tation as main driving input. For illustration purposes, two events of two-day duration each are shown. The first event occurred in spring (May 2010), the second in both events, and shows that the uncertainty is high when there is a high precipitation event. The more intense the precipitation input, as seen in the figure inset at top-left (May event), the greater the uncertainty band width for overflow flow, as well as for COD and $NH_4$ loads (insets three and four) and concentrations (insets five and bottom). 480 The MC estimated statistics and the standard errors (se) are presented in Table 7. The table shows the uncertainty quantification of outputs obtained from the MC uncertainty propagation for the first MC simulation (all selected input variables are uncertain).

**Table 7.** Monte Carlo estimated statistics and standard errors (se) by bootstrapping for the MC simulations, where all selected input variables are uncertain R (model run at 10 minute time steps and MC results aggregated to one hour averages over one year period). See Table 1 for output variable nomenclature and units. Interq. = Interquartile; quant. = quantile; pbw = prediction band width; sd = standard deviation; var = variance; cv = coefficient of variation.

| | Mean | Interq. Range | 0.005 quant. | 0.995 quant. | 99% pbw | 0.05 quant. | 0.95 quant. | 90% pbw | sd | var | cv |
|---|---|---|---|---|---|---|---|---|---|---|---|
| | $\mu_{MC}$ | IQR | $\zeta_{0.005}$ | $\zeta_{0.995}$ | $\zeta_{w,0.99}$ | $\zeta_{0.05}$ | $\zeta_{0.95}$ | $\zeta_{w,0.90}$ | $\sigma_{MC}$ | $\sigma^2_{MC}$ | $CV_{MC}$ |
| $V_{Chamber}$ | 92.51 | 13.65 | 77.55 | 109.16 | 31.60 | 81.02 | 104.28 | 23.25 | 8.27 | 2,984 | 0.100 |
| se | 2.53 | 1.17 | 2.18 | 3.17 | 1.48 | 2.27 | 2.95 | 1.36 | 0.56 | 482 | 0.001 |
| $V_{Sv}$ | 3.18 | 3.69 | 0.85 | 6.60 | 5.76 | 0.94 | 5.76 | 4.82 | 1.98 | 1,100 | 0.070 |
| se | 0.51 | 0.73 | 0.24 | 0.95 | 0.95 | 0.26 | 0.91 | 0.83 | 0.35 | 259 | 0.012 |
| $Q_{Sv}$ | 55.49 | 76.77 | 0.37 | 267.1 | 266.7 | 1.24 | 165.3 | 164.1 | 64.50 | 7,332 | 1.585 |
| se | 5.67 | 10.26 | 0.13 | 13.75 | 14.54 | 0.22 | 10.82 | 10.89 | 4.57 | 1,102 | 0.048 |
| $B_{COD,Sv}$ | 1.18 | 1.69 | 0.04 | 6.11 | 6.06 | 0.07 | 3.49 | 3.41 | 1.27 | 394 | 0.087 |
| se | 0.20 | 0.31 | 0.02 | 1.02 | 1.00 | 0.03 | 0.59 | 0.59 | 0.21 | 81.71 | 0.013 |
| $B_{NH4,Sv}$ | 0.052 | 0.077 | 0.004 | 0.174 | 0.170 | 0.006 | 0.125 | 0.120 | 0.045 | 0.546 | 0.075 |
| se | 0.009 | 0.014 | 0.002 | 0.029 | 0.028 | 0.002 | 0.022 | 0.021 | 0.008 | 0.115 | 0.013 |
| $C_{COD,Sv,av}$ | 170.0 | 164.6 | 3.80 | 909.7 | 905.9 | 15.49 | 465.8 | 450.3 | 161.9 | 36,151 | 0.988 |
| se | 9.02 | 11.38 | 0.33 | 40.12 | 40.40 | 1.02 | 24.66 | 24.03 | 7.94 | 4,615 | 0.016 |
| $C_{NH4,Sv,av}$ | 7.19 | 6.65 | 0.47 | 29.20 | 28.74 | 0.86 | 17.51 | 16.64 | 5.66 | 46.93 | 0.815 |
| se | 0.41 | 0.61 | 0.02 | 1.23 | 1.22 | 0.06 | 0.91 | 0.87 | 0.29 | 6.62 | 0.016 |





Table 7 shows the standard deviation (sd) and the coefficient of variation (cv) for the seven output variables considered in the uncertainty propagation. For the volume in the CSO chamber, $V_{Chamber}$, the annual mean standard deviation, $\sigma_{\mathrm{MC}}$, (8.27 m$^3$) is lower than the mean, $\mu_{\mathrm{MC}}$, (92.51 m$^3$). This goes along with an annual mean coefficient of variation ($CV_{\mathrm{MC}}$) of 0.100. A ($CV_{\mathrm{MC}}$) greater than 1 means large uncertainty. The overflow spill volume, $V_{Sv}$, had a coefficient of variation of 0.070, while it was 1.585 for the overflow flow, $Q_{Sv}$. This shows that the relative uncertainty of the overflow flow is very large. Regarding the overflow COD load, the annual mean (1.18 kg) is similar as the annual mean standard deviation (1.27 kg). Similar behaviour was observed for the overflow COD concentration, which had an annual mean value of 170 mg/l and a standard deviation of 162 mg/l. For overflow NH$_4$ load and overflow NH$_4$ concentration the annual mean also had the same order of magnitude as the annual mean standard deviation. Overflow COD and NH$_4$ loads had a coefficient of variation of 0.087 and 0.075, respectively, whereas the coefficient of variation for concentrations were 0.988 and 0.815, respectively. This suggests that overflow concentrations are more uncertain.

Low standard errors (se) for the coefficient of variation were obtained for all seven outputs. These were never greater than 0.05, which indicates that the selected MC replication size (1,500 for $mc1$) is a suitable value. This holds for all output statistics, because in all cases the standard error is small to the estimated value.

### 3.3.3 Contributions of input variables to total uncertainty

The contributions of input variables to the total uncertainty of the model inputs were also computed using the procedure described in Section 2.6.4. A total of four MC simulations with a total of 6,000 runs were performed for estimating $S_i$ (Eq. (16)). Afterwards, four contributions were evaluated per time step and aggregated for the whole year. Following Eq. (16), the per time step contributions of input variables to output variables in terms of percentage of variance, stochastic sensitivity $S_i$ of the input variables $C_{COD,S}$, $C_{NH4,S}$, $COD_r$ and $P$ were calculated. An example of the contributions analysis per time step is presented in Fig. 6. Here we remark that a high uncertainty over time is shown mainly for the Spring event.

The aggregated over time contributions of input variables to output variables in terms of percentage of variance, stochastic sensitivity $S_i$ of the input variables, were also calculated (Table 8). Note that $P$ is the only source of uncertainty for $V_{Chamber}$ and $V_{Sv}$, while uncertainty in NH$_4$ inputs only propagates to NH$_4$ outputs, and similar for COD (Fig. 6).

**Table 8.** Aggregated over time contribution of input variables to output variables in terms of percentage of total variance

| Output variable | Total | $C_{COD,S}$ | $C_{NH4,s}$ | $COD_r$ | $P$ |
|---|---|---|---|---|---|
| | | | Stochastic sensitivity, $S_i$, of input variable [%] | | |
| $V_{Chamber}$ | 100.0 | 0.0 | 0.0 | 0.0 | 100.0 |
| $V_{Sv}$ | 100.0 | 0.0 | 0.0 | 0.0 | 100.0 |
| $B_{COD,Sv}$ | 100.0 | 65.7 | 0.0 | 2.9 | 31.4 |
| $C_{COD,Sv,av}$ | 100.0 | 62.4 | 0.0 | 8.7 | 28.9 |
| $B_{NH4,Sv}$ | 100.0 | 0.0 | 34.4 | 0.0 | 65.6 |
| $C_{NH4,Sv,av}$ | 100.0 | 0.0 | 35.3 | 0.0 | 64.7 |





**Figure 5.** Uncertainty propagation outcomes for the first Monte Carlo simulation, where all input variables vary stochastically. The 99% prediction interval is shown as light grey shade, 90% prediction interval is shown as dark grey shade, mean value as blue line. The MC simulations were performed for the entire year 2010 at 10 minutes time step, aggregated to hourly time steps in the figure. Input precipitation (**top**). Overflow spill flow, the upper dashed red line indicates the 75 l/s threshold, lower dotted red line the 37.5 l/s threshold (**second**). Load of overflow COD (**third**). Load of overflow NH$_4$ (**fourth**). Average spill COD concentration. Upper dashed red line indicates the 125 mg/l threshold and the lower dotted red line indicates the 90 mg/l threshold (**fifth**). Average spill NH$_4$ concentration. Upper dashed red line indicates the 5.0 mg/l threshold, lower dotted red line the 2.5 mg/l threshold (**bottom**).

**Figure 6.** Temporal contributions of input variables to load of overflow COD (**top**) load of overflow NH$_4$ (**second**); concentration of overflow COD (**third**); concentration of overflow NH$_4$ (**bottom**) in terms of variance. The MC simulations were performed for the entire year 2010 at 10 minutes time step, which were aggregated to hourly time steps. For illustration two periods are shown from 28 to 30 May 2010 (**left**); and from 7 to 9 September 2010 (**right**).





We found, as expected, that precipitation, $P$, is the only source of uncertainty from all uncertain input considered for water quantity output variables $V_{Chamber}$ and $V_{Sv}$. Regarding average values for the whole year, for the water quality output variables $B_{COD,Sv}$ and $C_{COD,Sv,av}$, $C_{COD,s}$ has the largest contribution to the output variance, about 66 percent for $B_{COD,Sv}$ and about 62 percent for $C_{COD,Sv,av}$. The second variable that contributes to uncertainty of these COD output variables is $P$, with about 510    3 percent for $B_{COD,Sv}$ and 9 percent for $C_{COD,Sv,av}$. Similarly, the input variable $C_{NH4,S}$ play an important role in the contribution of total uncertainty for $B_{NH4,Sv}$ (on average about 34 percent of the variance for the whole year) and $C_{NH4,Sv,av}$ (about 35 percent). Equally contributing to uncertainty of these NH$_4$ output variables is $P$ with about 66 percent for $B_{NH4,Sv}$ and 65 percent for $C_{NH4,Sv,av}$. From these results we can infer that precipitation is a main source of uncertainty for all six outputs considered.

### 3.4    Uncertainty and water quality impact

Quantification and assessment of the water quality impact is an important step after the uncertainty propagation. As described in Section 2.7, the assessment of water quality standards was done taking into account the reference thresholds recommended in the European Union guidelines for COD, and the German and Austrian guidelines for hydraulic impact and acute ammonium toxicity.

### 3.4.1    Hydraulic impact

From the time series of daily values for 2006 to 2013 of the river Sûre, a daily flow expected with return period once per year (1.01 years), $Q_{r1}$ of 16 m$^3\cdot$s$^{-1}$ was computed at Heiderscheidergrund, which corresponds with the entire catchment area of the Haute-Sûre stormwater system (182.1 ha). Therefore, we estimated the river daily flow in the Goesdorf CSOT as a proportion to 30 ha, which is equal to 2.6 m$^3\cdot$s$^{-1}$. Following Eq. (17), the maximum sewer overflow discharge with return period one 525    year $Q_1$ can have a value between 0.26 m$^3\cdot$s$^{-1}$ and 1.32 m$^3\cdot$s$^{-1}$. Accordingly, with the German guideline ATV-A 128 (1992) (Eq. (18)), two additional thresholds are defined for the maximum sewer overflow discharge with return period one year for the Goesdorf catchment ($A_{imp} = 5.0$ ha). $Q_1$ is expected to vary between 37.5 l$\cdot$s$^{-1}$ and 75.0 l$\cdot$s$^{-1}$. We contrasted these values with those obtained from the uncertainty analysis. From Table 7, we obtained a one hour mean value for the overflow spill flow, $Q_{Sv}$, of 55.5 l$\cdot$s$^{-1}$, 90% prediction band width of 164.1 l$\cdot$s$^{-1}$, and standard deviation of 64.5 l$\cdot$s$^{-1}$. Figure 5 (second) presents 530    the overflow spill flow for the two periods chosen for illustration. The upper dashed red line indicates the 75 l$\cdot$s$^{-1}$ threshold and the lower dotted red line indicates the 37.5 l$\cdot$s$^{-1}$ threshold. Table 9 (top) shows the exceedance percentage of overflow spill flow over the 37.5 and 75.0 l$\cdot$s$^{-1}$ thresholds for the mean, 0.95 quantile and 0.995 quantile. We found a 0.49% exceedance of the mean value over the 37.5 l$\cdot$s$^{-1}$ threshold and about 1.7% for the quantiles. As expected, slightly lower percentages were found for the 75.0 l$\cdot$s$^{-1}$ threshold.





### 3.4.2 COD concentration

A reference COD concentration emission in CSOs was presented in Section 2.7.2. For the European Union, a value of 125 mg·l$^{-1}$ is used. We obtained a one hour average spill COD concentration with a mean of 170 mg·l$^{-1}$, standard deviation of 162 mg·l$^{-1}$, and a 90% prediction band width of 450 mg·l$^{-1}$. Figure 5 (fifth) presents the average spill COD concentration. Upper dashed red line indicates the 125 mg/l threshold, lower dotted red line the 90 mg/l threshold. The mean COD concentration

in the overflow volume was higher than the thresholds. However, note that when entering the river system it will quickly be diluted, suggesting that the negative impact on the environment will be dampened by the receiving water body.

Table 9 (centre) shows the exceedance percentage of overflow COD concentration over the 90 and 125 mg·l$^{-1}$ thresholds for the mean, 0.95 quantile and 0.995 quantile. We found a 1.62% exceedance of the mean value over the 90 mg·l$^{-1}$ threshold and about 1.8% for the quantiles. Slightly lower percentages were found for the 125 mg·l$^{-1}$ threshold for the mean value (1.03%).

For the quantiles equal values were found as for the 90 mg·l$^{-1}$ threshold.

### 3.4.3 Acute ammonium toxicity

We compared the acute ammonium toxicity reference values presented in Section 2.7.3 (2.5 mg·l$^{-1}$ for the ammonium concentration calculated for one hour duration for salmonid streams, and for cyprinid streams a maximum value of 5.0 mg·l$^{-1}$), with the values we found for ammonium. An average spill NH$_4$ concentration with a mean of 7.19 mg·l$^{-1}$, standard deviation

of 5.66 mg·l$^{-1}$, and 90% prediction band width of 16.64 mg·l$^{-1}$ was obtained. Figure 5 (bottom) shows the average spill NH$_4$ concentration for the two periods chosen for illustration. The ammonium (NH$_4$) concentrations in the overflow flow are higher than the reference values, which are given for concentrations in the river.

Table 9 (bottom) shows the exceedance percentage of overflow NH$_4$ concentration over the 2.5 and 5.0 mg·l$^{-1}$ thresholds for the mean, 0.95 quantile and 0.995 quantile. We found a 1.8% exceedance of the mean and quantile values over the 2.5 and

5.0 mg·l$^{-1}$ thresholds. A slightly lower percentage (1.1%) was found for the 5.0 mg·l$^{-1}$ threshold, regarding mean value.

### 4  Discussion

This study aimed to select and characterise the main sources of input uncertainty in urban water systems, while accounting for temporal auto- and cross-correlation of uncertain model inputs, by propagating input uncertainty through the EmiStatR model, and quantifying and assessing the contributions of each uncertainty source to model output uncertainty dynamically (over time).

In the following discussion, we start with the accuracy of Monte Carlo analysis. Then, we discuss the water quality impact of the model outputs to the environment, in relation to the uncertainty analysis, and finally, we highlight some limitations and possible solutions of the approach used in this work.





**Table 9.** Frequency (percentage) over time that environmental thresholds are exceeded for different statistics of the overflow spill flow, COD and $NH_4$ concentration.

| Output variable | Threshold | Statistic | Exceedance percentage |
|---|---|---|---|
| $Q_{Sv}$ [l·s$^{-1}$] | 37.5 | Mean | 0.49 |
| | 37.5 | 0.95 quantile | 1.71 |
| | 37.5 | 0.995 quantile | 1.74 |
| | 75.0 | Mean | 0.31 |
| | 75.0 | 0.95 quantile | 1.51 |
| | 75.0 | 0.995 quantile | 1.72 |
| $C_{COD,Sv,av}$ [mg·l$^{-1}$] | 90.0 | Mean | 1.62 |
| | 90.0 | 0.95 quantile | 1.80 |
| | 90.0 | 0.995 quantile | 1.82 |
| | 125.0 | Mean | 1.03 |
| | 125.0 | 0.95 quantile | 1.80 |
| | 125.0 | 0.995 quantile | 1.82 |
| $C_{NH4,Sv,av}$ [mg·l$^{-1}$] | 2.5 | Mean | 1.78 |
| | 2.5 | 0.95 quantile | 1.80 |
| | 2.5 | 0.995 quantile | 1.82 |
| | 5.0 | Mean | 1.05 |
| | 5.0 | 0.95 quantile | 1.78 |
| | 5.0 | 0.995 quantile | 1.82 |

## 4.1 Uncertainty and water quality impact

Next we discuss how the uncertainty propagation analysis done gives additional insight regarding hydraulics, COD concentra-
tion and acute ammonium toxicity impact on water quality over the river Sûre due to the CSO discharges under study. After
doing the uncertainty propagation analysis we not only have predictions of model outputs but we also know how uncertain
these are. An added value arises when we take into account the uncertainty information. For the case of the overflow spill flow,
the expected model output (mean of 55.5 l·s$^{-1}$) is below the environmental threshold of 75 l·s$^{-1}$, but the 0.95 quantile (164.1
l·s$^{-1}$) is much above the threshold. This indicates that there is a considerable chance of being above the threshold.
Regarding water quality outputs, although the mean model output for COD and $NH_4$ concentrations is fairly above of the
thresholds, the 0.95 quantile is 2.7 times above the mean value for COD concentration, and 2.4 times above the mean value
for $NH_4$. Also here we can conclude that we are not certain that we are below the threshold, because there is a considerable
probability that the true values are above, even though the expected value is below the thresholds.





We were able to compute the water quantity and quality at CSO outlet to the river. We found that water quality (COD and
NH$_4$) were sometimes above the environmental threshold. Even if the expected value was below the threshold there could still
be a considerable probability that the quality was above the threshold because of the large uncertainty. Therefore, policy and
decision makers and water managers need to be aware of this, because whenever concentrations are above the threshold this
may harm the environment. Nevertheless it is worth noting that we computed concentration in the outlet of the CSO. When this
spilled water enters the river it will quickly mix with the much cleaner river water and concentrations will drop quickly, so it
is only a local problem. How local it is and how the river water quality is distributed in space and time is not an easy problem
to solve and requires the use of hydrological and hydraulic river models e.g. SIMBA (IFAK, 2007) or MIKE 11 (DHI, 2017).
Those models have been well-developed and for some of them uncertainty analyses have also been done (Beven and Binley,
1992; Refsgaard, 1997; Beven and Freer, 2001; Vrugt et al., 2003a, 2008; Beven et al., 2010; Andrés-Doménech et al., 2010;
Beven, 2012; Jerves-Cobo et al., 2020; Yu et al., 2020), but obviously such uncertainty analyses can only be done if the inputs
to these models are known as well as the uncertainty associated with these inputs. One of these inputs is inlet from CSO. That
is where our paper makes a very valuable contribution, because our work has quantified water quantity and quality of CSO
structures, including uncertainty, and that is exactly what these river models need to be able to do an uncertainty propagation
analysis.

## 4.2  Accuracy of Monte Carlo analysis

Regarding the Monte Carlo replication size for uncertainty propagation, we presented in Fig. 4 the results for three output
variables and three replications size 250, 1,000 and the selected 1,500 (NSE closer to 1.0 for most of the output variables). We
compute replications for 50, 100, and from 250 to 2,000 at steps of 250 replications the comparison of two equal MC runs (MC1
and MC2) with different seed for the pseudo-random number generator. the results suggest that the output variables related to
COD (load and concentration) have a larger dispersion when we compare MC1 and MC2 for the same replications size. This
is also reflected in the larger standard errors reported in Table 7 for e.g. the overflow COD load. Nevertheless, 1,500 runs are a
feasible MC replication size for running a relative simple and fast model as EmiStatR (7.29 minutes in average execution time
using parallel computing and 50 cores for a time series with 4,464 time steps). For a more complex full hydrodynamic model
with a high computational burden, 1,500 replication four times to compute contributions it may be not possible. Therefore,
we suggest to check the intermediate results of the MC convergence test and we will find that e.g. for quantity variables as
the spill overflow volume and quality variables as the overflow NH$_4$ load, 250 replications (7.10 minutes in average execution
time using parallel computing and three cores for a time series with 4,464 time steps) per individual MC execution seem to be
enough, which make more feasible the execution of this kind of uncertainty propagation.

Figures 5 and 6 shows that there is a large uncertainty for the early May event and smaller uncertainty for the September
event. This is due mainly to the presence of a large dry period before the spill event in May, i.e. a shorter dry period preceding
the spill flow leads to a lower uncertainty. This finding suggest also an interaction between the antecedent dry period and the
concentration of pollutants.





### 4.3 Other sources of uncertainty

In this work we only looked at input uncertainty and not at parameter and model structural uncertainty. Further research can be done on those topics. Neumann (2007) address how are uncertainty ranges for parameters of full scale systems obtained and how does model structure uncertainty manifest itself and can be quantified for performance evaluation and design of urban water infrastructure. Moreno-Rodenas et al. (2019) also studied and depicted how model parameter is an important source of uncertainty. They emphasised that "*still, uncertainty analysis is seldom applied in practice and the relative contribution of the individual model elements is poorly understood.*". Also, they highlighted that after inferring the river process parameters with system measurements of flow and dissolved oxygen, combined sewer overflow pollution loads became the dominant uncertainty source along with rainfall variability. These findings agreed with our results.

Bachmann-Machnik et al. (2018) recognised that the most important parameters causing uncertainties in the sewer system model are connected area and the stormwater runoff quality. Our analysis confirms these findings, specifically regarding the stormwater runoff quality. In our study the input variable runoff COD was an important source of uncertainty with relation to the annual mean overflow COD released from the CSO.

### 4.4 Limitations and possible improvements

Despite the extensive temporal uncertainty propagation analysis the approach also has some limitations which we present hereafter addressing possible solutions in future work.

1. **Incorporation of the spatial distribution of model inputs.** Specifically for precipitation, Breinholt et al. (2012) stated that due to a poor representation of the spatial precipitation that is measured by point gauges and the complexity of the sewer systems, large output uncertainty can be expected. We also infer that we obtained a large output uncertainty due to neglect of the inherent spatial variability of precipitation. Therefore, we suggest that further research is needed to account for spatial variability of precipitation, that can bring light to understand how this variability impacts in the output uncertainty and quantify it properly. This issue should be related also to the problem of change of support. When modelling precipitation, we also ignored the support effect, i.e. we ignored that the sub-catchment area is much greater than a point. Future research may address this issue of change-of-support. Studies that tackled this issue are found e.g. Leopold et al. (2006); Wadoux et al. (2017); Cecinati et al. (2018).

2. **Linkage of sub-models and uncertainty compensation effect.** Tscheikner-Gratl et al. (2019) addressed the question as to whether there is an increase in uncertainty by linking integrated models or if a compensation effect could take place and that overall uncertainty in key water quality parameters actually decreases. We contribute in this discussion by advising to quantify uncertainties at sub-model level, because as we demonstrated the computational budget can be reduced and make it feasible when dealing at the sub-module uncertainty propagation.

3. **Accounting for cross-correlation between the inputs precipitation and runoff COD concentration.** It is worth noting that we did not include correlation between $COD_r$ and $P$. Including such correlation would yield a more realistic





model of the uncertainty because these variables are known to have a strong correlation. It is highly recommendable to
include correlations between $COD_r$ and precipitation, because loads in chemical oxygen demand are correlated with
the overland flow due to precipitation, which may transport distributed pollutants to the sewer system. Also the inputs
$C_{COD,s}$ and $C_{NH_4,s}$ can be related with a daily curve that reflect the pattern of consumption in the household like
the German ATV-A 134 curve. We used the latest version of EmiStatR (version 1.2.2.0), which considers this kind of
patterns.

4. **Absence of high frequency water quality observations to compare with model outputs and uncertainty prediction
bands.** In order to gain understanding of the temporal dynamics of nutrients (nitrogen, N, and phosphorus, P), Yu et al.
(2020) applied high frequency monitoring in a groundwater fed low-lying urban polder in Amsterdam (The Netherlands).
They argued that although spatial and temporal concentration patterns from discrete sampling campaigns of water quality
parameters, such as EColi, showed a clear dilution pattern, the temporal patterns of N and P were still poorly understood,
given their reactive nature and more complex biogeochemistry. Therefore, high frequency measurement, is a key factor
to understand these temporal dynamics and patterns.

5. **Absence of a joint spatio-temporal uncertainty analysis.** According to Zhou et al. (2020), the limitations in algorithms
for classic uncertainty estimates is the cause that only the uncertainty in one dimension (either temporal variability or
spatial heterogeneity) is considered, whereas the variation in the other dimension is dismissed, resulting in an incomplete
assessment of the uncertainties. Zhou et al. (2020), also showed that classic metrics underestimate the uncertainty through
averaging, which means a loss of information in the variation across spatio-temporal scales. To handle this limitation,
suitable methods are the three-dimensional variance partitioning for a new uncertainty estimation in both spatio-temporal
scales (Zhou et al., 2020), or spatio-temporal geostatistics (Gräler et al., 2016).

## 5 Conclusions

In this final section we conclude with highlighting the importance of temporal uncertainty propagation analysis and the selec-
tion and characterisation of uncertain model inputs impacting model sensitivity. We also point out that uncertainty propagation
analysis helps to identify the most contributing sources and can provide better evidence for the impact assessment of pollutant
release from sewer systems to the environment, in particular to the receiving waters.

1. **Uncertainty analysis is important because it quantifies the accuracy of model outputs and quantifies the uncer-
tainty source contributions.** The latter provides essential information to take informed decisions about how to improve
the accuracy of the model output. But MC uncertainty analysis is only possible if it is computationally feasible. We used a
simplified urban water system model with capabilities to apply for *minimising transient pollution from urban wastewater
systems* in parallel mode, which minimises model running time, allowing uncertainty propagation, long term simulations
and evaluation of complex scenarios. These capabilities are crucial also for e.g. real time control applications, where
simplified models of fast running times are desirable.





2. **Input variables that were very uncertain for which model output was very sensitive were selected to be included in the uncertainty propagation analysis.** We found four main input variables to be analysed: 1) Precipitation, $P$; 2) Chemical oxygen demand sewage pollution per capita load per day, $C_{COD,S}$; 3) Ammonium pollution per capita load per day, $C_{NH_4,S}$; and 4) Chemical oxygen demand $COD_r$ concentration.

3. **Selected input variables for uncertainty propagation can be characterised in terms of input uncertainty in four specific cases, depending on the type of input variable:** i) Uncertain constant inputs, characterised by their marginal (cumulative) pdf e.g. water consumption, infiltration flow, impervious area and run-off coefficients; ii) Temporally autocorrelated dynamic uncertain inputs, characterised by univariate time series autoregressive modelling e.g. $COD_r$; iii) Temporally cross-correlated multiple dynamic uncertain inputs, characterised by multivariate time series modelling, considering cross- and no-correlations among variables e.g. $C_{COD,S}$ and $C_{NH_4,S}$; and iv) rain gauge input precipitation, characterised by autoregressive model conditioned to the observed precipitation ($P$).

4. **Model input uncertainty propagation through the simplified combined sewer overflow model (EmiStatR) helped to understand how does uncertainty propagate and how large is the uncertainty of EmiStatR outputs in a case study.** Three output variables were considered for water quantity and four variables for water quality. The Monte Carlo uncertainty propagation analysis showed that among the water quantity output variables, the overflow flow, $Q_{Sv}$, is the more uncertain output variable and has a large coefficient of variation (cv of 1.585). Among water quality variables, the annual average spill COD concentration, $C_{COD,Sv,av}$, and the average spill NH$_4$ concentration, $C_{NH_4,Sv,av}$, were found to have large uncertainty (coefficients of variation of 0.988 and 0.815, respectively). Also, low standard errors (se) for the coefficient of variation were obtained for all seven outputs. They were never greater than 0.05, which indicated that the selected MC replication size (1,500 simulations) was a suitable value.

5. Regarding the **main sources of uncertainty model outputs**, for water quantity outputs, was precipitation, while for COD water quality outputs were $P$, $C_{COD,S}$ and $COD_r$, and for NH$_4$ outputs $P$ and $C_{NH_4,S}$.

6. Finally, we evaluated how **uncertainty propagation analysis can explain more comprehensively the impact of water quality indicators to the receiving river** for the Luxembourg case study. Although the mean model water quality outputs for COD and NH$_4$ concentrations is fairly above of the thresholds, the 0,95 quantile is 2.7 times above the mean value for COD concentration, and 2.4 times above the mean value for NH$_4$. We conclude that we are not certain that environmental thresholds are not exceeded, because there is a considerable probability that values are above, even though the expected value is below the thresholds. This is valid for concentrations in the spilled CSO, therefore, is important to highlight that the results confirmed our hypothesis that annual mean COD and NH$_4$ river concentrations are lower than the released CSO concentrations due to dilution and henceforth compliant with the water quality thresholds given by the guidelines consulted.

*Code and data availability.*  The code scripts and datasets related to Figures 03 to 06 of this paper are available on Zenodo:





https://doi.org/10.5281/zenodo.3928079

and GitHub:

https://github.com/ArturoTorres/temporal_uncertainty_paper_reproducible.git

*Author contributions.*   J. A. Torres-Matallana is the main author of the text in this article. Ulrich Leopold and Gerard Heuvelink contributed to this article with their statistical, geostatistical and programming knowledge, and reviewed and edited the text. J. A. Torres-Matallana developed all R-code scripts for the computations and Monte Carlo simulations, performed the simulations and analysis, with collaboration from Ulrich Leopold and Gerard Heuvelink.

*Competing interests.*   The authors declare no competing interests.

*Acknowledgements.*   The work presented was part of the QUICS (Quantifying Uncertainty in Integrated Catchment Studies) project. This project has received funding from the EU Marie Skłodowska-Curie research programme under the European Union's Seventh Framework Programme for research, technological development and demonstration with the grant agreement No. 607000, as well as the Luxembourg Institute of Science and Technology, LIST. We thank Dr. Kai Klepiszewski for his advice and involvement during early stages of this research.





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
