# Peer review of "Multivariate autoregressive modelling and conditional simulation for temporal uncertainty propagation in urban water systems"

_Hydrology and Earth System Sciences, 2020_

## Referee Comment (RC1) · Anonymous Referee #1 · 31 Aug 2020

The manuscript "Multivariate autoregressive modelling and conditional simulation for temporal uncertainty propagation in urban water systems" aims to select and characterize the main sources of input uncertainty in urban water systems and quantifying the contributions of each uncertainty source to model output uncertainty over time. It provides a good and well-structured example of an uncertainty analysis for a quite simplified model. Especially the results on CSO water quantity and quality are interesting and useful for further studies. In general, I think the manuscript is a valuable addition to the field with some minor discussion points I would like to bring up:

[Figure]

Page 2 – Line 40: Is the minimization of CSO volume alone a goal in itself? There is the question if many events with a bad water quality (e.g. first flush) are better than fewer events with higher volume and better water quality? That may be a point for elaboration.

Page 12 – Line 261: As your model is quite simple and requires "little" computational time the chosen method is feasible, but that is not the case in most of those integrated studies. Is that not a limitation worth mentioning and discussing in 4.4? How could the approach look like in a more complex model?

Page 19 – Table 4: I quite like this very accessible and clear table for the decision-making of which input variables you select. Still, I think that the variables that are awarded ++ and + for uncertainty and sensitivity respectively must be discussed more. Especially I think that on the infiltration, NH4 in Rainwater, and C pervious where I don't necessarily agree with omitting them, at least not on the argues in the text of 3.1. On the other hand, I am surprised on the uncertainty of the total area. So, the distinction where the authors draw the deciding line in what to include into their analysis must be clearer. It could be maybe better explained by using graphical panels (e.g. in QUICS (Tscheikner-Gratl et al., 2017)) for illustrating that decision.

Page 29 – Line 560: You don't start with the accuracy of Monte Carlo Analysis (which is then 4.2) but with Uncertainty and water quality impact (4.1).

Page 31 – Line 588: I agree that that is one of the very valuable contribution of this paper. Still I would like to see some comparisons to other attempts on quantity (e.g. Sriwastava et al., 2018) and quality (especially measurements taken at CSOs the measured water quality at the WWTP influent is expected to render a low representativity of the conditions at the CSOs - e.g. Brombach et al.(2005); Diaz-Fierros T et al. (2002))

Page 32 – Line 62: The point about linkage is an important one, but I don't see the big input from this paper on the topic. Can you elaborate on this, why is the quantification at sub-module level advisable? Only due to the computational budget limitations?
Page 34 - Line 701: Your abstract starts with "Uncertainty is often ignored in urban water systems modelling." I would have therefore expected and would like to read how this can be improved and how studies like yours can provide guidance for the decision makers.

Literature: Brombach, H., Weiss, G., Fuchs, S., 2005. A new database on urban runoff pollution: comparison of separate and combined sewer systems. Water Sci Technol 51, 119–128. https://doi.org/10.2166/wst.2005.0039 Diaz-Fierros T, F., Puerta, J., Suarez, J., Diaz-Fierros V, F., 2002. Contaminant loads of CSOs at the wastewater treatment plant of a city in NW Spain. Urban Water 4, 291–299. https://doi.org/10.1016/S1462-0758(02)00020-1 Sriwastava, A.K., Tait, S., Schellart, A., Kroll, S., Van Dorpe, M., Van Assel, J., Shucksmith, J., 2018. Quantifying Uncertainty in Simulation of Sewer Overflow Volume. Journal of Environmental Engineering 144, 04018050. https://doi.org/10.1061/(ASCE)EE.1943-7870.0001392 Tscheikner-Gratl, F., Lepot, M., Moreno-Rodenas, A., Schellart, A., 2017. A Framework for the application of Uncertainty Analysis (Deliverable No. 6.7), QUICS. Zenodo, https://zenodo.org/record/1240926

---

## Author Comment (AC1) · 15 Sep 2020

Thank you for your kind words and valuable comments that helped us to improve the manuscript. We considered each comment and you can find our replies below after each comment from the Referee #1.

[1] Page 2 – Line 40: Is the minimization of CSO volume alone a goal in itself? There is the question if many events with a bad water quality (e.g. first flush) are better than fewer events with higher volume and better water quality? That may be a point for

[Figure]

elaboration.

Reply: Good point. Minimization of CSO volume is not the only goal. In the revision, we will change the sentence to: "To reduce pollution in receiving waters it is important to minimise CSO load and concentration".

[2] Page 12 – Line 261: As your model is quite simple and requires "little" computational time the chosen method is feasible, but that is not the case in most of those integrated studies. Is that not a limitation worth mentioning and discussing in 4.4? How could the approach look like in a more complex model?

Reply: We agree that this point should be addressed in the Discussion and we will make appropriate adaptations in the revised manuscript in Section 4.4:

6. Uncertainty analysis with complex models. In this research, we were able to conduct a comprehensive Monte Carlo uncertainty propagation analysis, which required a large number of Monte Carlo runs. This was possible because we used a strongly simplified urban water system model, EmistatR. For more complex models that take much more computing time, application of a Monte Carlo uncertainty propagation analysis is more challenging. However, given sufficient resources, it is possible, because each model run can be run independently and hence the analysis is extremely suitable for parallisation and cloud computing. In particular, the use of graphics processing units (GPU) for heavy computation is promising. Some recent examples that demonstrate the potential of GPU for this purpose are Eranen et al. (2014), Sten et al. (2016) and Sandric et al. (2019). Sriwastava et al. (2018) applied uncertainty propagation to a complex hydrodynamic model, by selecting a small subset of dominant input/model parameters that explain most of the model output variance.

The methodology used in our study may be replicated for a model of higher complexity because of the scalable approach that was followed. The main limitation of application to a higher model complexity case is not the method implementation itself but the hardware setup that is required to make the uncertainty propagation feasible. It is necessary to speed up the computations of a single model run, which is not always an easy task.

[3] Page 19 – Table 4: I quite like this very accessible and clear table for the decision-making of which input variables you select. Still, I think that the variables that are awarded ++ and + for uncertainty and sensitivity respectively must be discussed more. Especially I think that on the infiltration, NH4 in Rainwater, and C pervious where I don't necessarily agree with omitting them, at least not on the argues in the text of 3.1. On the other hand, I am surprised on the uncertainty of the total area. So, the distinction where the authors draw the deciding line in what to include into their analysis must be clearer. It could be maybe better explained by using graphical panels (e.g. in QUICS (Tscheikner-Gratl et al., 2017)) for illustrating that decision.

Reply: We agree that the decision between ++ and + for uncertainty and sensitivity needs more justification and we will make appropriate adaptations in the revised manuscript for inflow of infiltration water, NH4 in rainwater, and C pervious:

Adaptation Section 3.1.2: Regarding the inflow of infiltration water (4), although this is a very uncertain input, the quick-scan analysis showed that model output sensitivity is not very high as is indicated in Table 3. For this reason, we do not include this variable in the uncertainty propagation analysis.

Adaptation Section 3.1.3: Regarding NH4 in rainwater (9), although model output is very sensitive to this model input variable, model input uncertainty is not very high as is indicated in Table 3. For this reason, it was not included in the uncertainty propagation analysis.

Adaptation Section 3.1.4: Although model output is very sensitive to the input variable Cper (13), the uncertainty about this variable is not very high, as indicated in Table 3. The reason behind this is that Cper can be derived fairly accurately from GIS products, such as land use and soil type maps. Therefore, we did not include this variable in the uncertainty propagation analysis.

To better support our decisions we will also include a figure as in Tscheikner-Gratl et al. (2017), as suggested by the reviewer (see below Fig. 1). We will either include this figure in the revision or in the Supporting Information.

[4] Page 29 – Line 560: You don't start with the accuracy of Monte Carlo Analysis (which is then 4.2) but with Uncertainty and water quality impact (4.1).

Reply: Thank you for noting this mistake. We will correct this in the revised manuscript, by changing the text to: "In the following discussion, we start with the uncertainty and water quality impact of the model outputs to the environment, in relation to the uncertainty analysis. Next, we discuss the accuracy of Monte Carlo analysis, followed by a discussion of other sources of uncertainty. Finally, we highlight some limitations and possible solutions of the approach used in this work."

[5] Page 31 – Line 588: I agree that that is one of the very valuable contribution of this paper. Still I would like to see some comparisons to other attempts on quantity (e.g. Sriwastava et al., 2018) and quality (especially measurements taken at CSOs the measured water quality at the WWTP influent is expected to render a low representativity of the conditions at the CSOs - e.g. Brombach et al.(2005); Diaz-Fierros T et al. (2002))

Reply: Thank you for your kind words and suggestion. We will make appropriate adaptations in the revised manuscript by expanding the text and including comparisons with other quantity and quality studies as follows:

"Sriwastava et al. (2018) apply uncertainty propagation to a complex hydrodynamic model for quantifying uncertainty in sewer overflow volume. They used MC for uncertainty propagation and Latin hypercube sampling (LHS) as an efficient sampling scheme. Although LHS ensures full coverage of the sample space and provides faster convergence than simple random sampling, the LHS application in the case of dynamic model inputs (e.g. precipitation, COD and NH4 inputs) is not trivial and its implementation is more complex than in the case of sampling from static variables (i.e., uncertain constants). In our study, we sampled time series of dynamic inputs using an implementation in stUPscales (Torres-Matallana et al., 2019; Torres-Matallana et al., 2018b).

Diaz-Fierros et al. (2002), in a study in the city of Santiago de Compostela (North-West Spain, population about 100,000 inhabitants), where a combined sewer system feeds to a grossly under-sized wastewater treatment plant, reported an event mean concentration (Diaz-Fierros et al. (2002), Table 4) for the output variables CCOD,Sv,av and CNH4,Sv,av of 329.1 mgÂůl−1 and 8.7 mgÂůl−1, respectively. These values are larger than those found by Brombach et al. (2005), and more in agreement with our findings, especially for the case of CNH4,Sv,av. Diaz-Fierros et al. (2002) reported values of CCOD,Sv,av as high as 1073 mgÂůl−1, which agrees with the right-hand tail of the distribution obtained in our study (i.e. a 0.995 quantile of 909.7 mgÂůl−1). Similarly, for the case of CNH4,Sv,av, Diaz-Fierros et al. (2002) reported values as high as 32.5 mgÂůl−1, comparable with the 0.995 quantile (29.20 mgÂůl−1) found in our study.

It is worth noting that regarding measurements taken at CSOs, the measured water quality at the WWTP influent is expected to render a low representativity of the conditions at the CSOs as reported by Diaz-Fierros et al. (2002) and Brombach et al. (2005). Thus, when comparing model outputs with independent measurements, one should bear in mind that discrepancies between measured and predicted are not only caused by errors in model inputs, model parameters and model structure but are also the result of errors in the water quality measurements."

[6] Page 32 – Line 62: The point about linkage is an important one, but I don't see the big input from this paper on the topic. Can you elaborate on this, why is the quantification at sub-module level advisable? Only due to the computational budget limitations?

Reply: We agree that we did not address this aspect in our paper but in the Discussion, we did want to point to the possibility of obtaining uncertainties at sub-model level. Some users may be interested at uncertainty levels of sub-modules of the model. For example, sub-module outputs are of particular interest in Bach et al. (2015), Burger et

al. (2016) and Rauch et al. (2017).

Perhaps the text on lines 632-636 was not very clear. We will reformulate it to: "Tscheikner-Gratl et al. (2019) addressed the question as to whether there is an increase in uncertainty by linking integrated models or whether a compensation effect could take place by which overall uncertainty in key water quality parameters decreases. Some further insight into this topic could be obtained by quantifying uncertainties at sub-model level, and analysing whether uncertainty at sub-model level is greater or smaller than at the overall level. With our implementation, this is not a difficult task because EmistatR has a stringent modular design in which it is easy to analyse outputs and their uncertainties at sub-model level."

[7] Page 34 - Line 701: Your abstract starts with "Uncertainty is often ignored in urban water systems modelling." I would have therefore expected and would like to read how this can be improved and how studies like yours can provide guidance for the decision-makers.

Reply: We believe that we have made a contribution towards making uncertainty propagation analysis in urban water systems modelling more routine. Clearly a single journal publication is not enough but we provide guidance, a simplified model that is very suited for Monte Carlo uncertainty propagation, and we shared the code scripts as well as the datasets to reproduce Figures 3 to 6, so that interested parties could more easily run an uncertainty analysis themselves. Please also note that our study was part of the larger 'QUICS' EU project (https://www.sheffield.ac.uk/quics), which aimed to stimulate the use of uncertainty analysis in integrated catchment modelling, and which involved partners from industry, water management authorities and consultancy firms.

Literature:

Brombach, H., Weiss, G., Fuchs, S., 2005. A new database on urban runoff pollution: comparison of separate and combined sewer systems. Water Sci Technol 51, 119–128. https://doi.org/10.2166/wst.2005.0039

Diaz-Fierros T, F., Puerta, J., Suarez, J., Diaz-Fierros V, F., 2002. Contaminant loads of CSOs at the wastewater treatment plant of a city in NW Spain. Urban Water 4, 291–299. https://doi.org/10.1016/S1462-0758(02)00020-1

Sriwastava, A.K., Tait, S., Schellart, A., Kroll, S., Van Dorpe, M., Van Assel, J., Shucksmith, J., 2018. Quantifying Uncertainty in Simulation of Sewer Overflow Volume. Journal of Environmental Engineering 144, 04018050. https://doi.org/10.1061/(ASCE)EE.1943-7870.0001392

Tscheikner-Gratl, F., Lepot, M., Moreno-Rodenas, A., Schellart, A., 2017. A Framework for the application of Uncertainty Analysis (Deliverable No. 6.7), QUICS. Zenodo, https://zenodo.org/record/1240926
* * *
[Figure]

**Fig. 1.** Graphical assessment of the contribution of input uncertainty to model output uncertainty. Numbers near each dot refer to the input variable number as defined in Table 4 of the manuscript.

---

## Referee Comment (RC2) · Jairo Arturo Torres-Matallana et al. · 16 Sep 2020

* General

This manuscript presents a detailed case study on uncertainty propagation through a water quality model. The authors propose the use of auto-regressive models to describe the dynamic of the input time-series.

For the reader it is currently unclear if the focus is on the method ("this paper introduces an uncertainty analysis framework") or the application case. My suggestion is to focus on the case study, as the methodological contribution is rather limited. In any case, the

focus should be set clearer in the introduction. In general, focusing more on the key points by moving some material to the supporting information would help.

The authors use AR(1) processes to model dynamical inputs. However, it needs also to be shown that these models captures the characteristics of the inputs correctly, for example that the auto-correlation function and other statistics match.

'Uncertainty analysis' is an umbrella term. Therefore I would encourage to use always the more specific terms 'uncertainty propagation' and 'sensitivity analysis' (SA) when referring to these concepts. SA is a well established term for calculating the "contributions of input variables to total uncertainty".

While hinted a several places, I think it would be beneficial to distinguish the kind of uncertainty that one tries to model more explicitly. Some inputs are intrinsically stochastic (e.g. precipitation), while others the uncertainty expresses our lack of knowledge about a parameter value.

* Specific points

** Abstract L4: the paper does not introduce a framework (which would be a theory about how to deal with uncertainties).

** Introduction

L54: "determinism" is the absence of uncertainty, hence it cannot "represent uncertainty"

** Material and methods

L115, eq2: Maybe I missed something, but is there no delay? The rain is transformed immediately to runoff?

L160: "Some variable" - which one?

Table 1: I would remove all irrelevant "inputs", such as ID, name of structure, ...

Table 2: What are "flow time structure" and "curve level"? They do not show up in the model equations

L165: This is an example where "uncertainty propagation" should be used instead of "uncertainty analysis"

L165ff: Why was the selection of inputs needed at all for the uncertainty propagation? The computational efforts do not change with the number of inputs (they do for the SA). Also there is not needed that the model is sensitive to an input. If it is not, than we will just get smaller uncertainties.

L179: the equation is for the cdf, not the pdf. Also emphasize on 'marginal' implies that you want to model a joint distribution.

L188ff: how did you determine the order of the AR process (seems always to be 1)?

L199: Not the uncertainties are correlated, but the values themselves

L211: Precipitation lot of zeros. It is not clear how the presented model can describe the dry periods.

L215-255: Remove the description as the model is already described in Torres-Matallana et al. (2017)

L257-260: Remove or move to introduction.

L 304: This is sensitive analysis and does not belong under the section "uncertainty propagation"

L317: How did you aggregate for the whole year? Is it a problem that the individual indices are not independent due to the auto-correlation?

L340: "may be computed" - did you do so?

** Results

L397: The wording is a bit confusing here: "evaluation of model output sensitivities"

sounds like a SA, but you are referring to the "manual" analysis of the model.

L415: was the COD modeled independently of the precipitation?

Figure3: Please mention in the caption which density you used of the uncertainty propagation. Also, maybe move figure to supporting information (SI).

L435: Please show some evidence for that.

L444-465: I propose to move this and figure 4 to the SI

** Discussion

L560: "we start with the accuracy..." - this topic comes second.

L637: This is an important suggestions. I'm a bit confused why you did not considered the correlation if the model is apparently able to do so ("We used the latest version of EmiStat R (version 1.2.2.0), which considers this kind of patterns.")
* * *

---

## Author Comment (AC2) · 7 Oct 2020

Reply: We welcome your comments, which were very helpful to improve our manuscript. In the text below we provide our replies (in blue) after each original comment.

\* General

This manuscript presents a detailed case study on uncertainty propagation through

a water quality model. The authors propose the use of auto-regressive models to describe the dynamic of the input time-series.

For the reader it is currently unclear if the focus is on the method ("this paper introduces an uncertainty analysis framework") or the application case. My suggestion is to focus on the case study, as the methodological contribution is rather limited. In any case, the focus should be set clearer in the introduction. In general, focusing more on the key points by moving some material to the supporting information would help.

Reply: We agree that the added value of the paper is mainly in the application, and we will make this clear in the revision, for example by no longer claiming that we introduce an uncertainty analysis framework but instead recognise that we build on existing methods. To emphasise that the application is most important we also changed the title of the manuscript to "Multivariate autoregressive modelling and conditional simulation for temporal uncertainty analysis of an urban water system **in Luxembourg**".

Even though the focus is on the application we do think that our paper also makes a valid contribution on methods. For example, we use both univariate and multivariate autoregressive models to characterise uncertainty in dynamic variables, we use conditional simulation to sample from these models, and we use bootstrap computation to summarise the Monte Carlo outputs. None of these methods are new but as far as we know they have not been used jointly in uncertainty analysis of urban water systems. We will move some of the materials to the supporting information, in particular the second part of Section 2.5.4 where we explain the calibration and conditional simulation of the precipitation time series (i.e., lines 223 to 255 in the original manuscript).

The authors use AR(1) processes to model dynamical inputs. However, it needs also to be shown that these models captures the characteristics of the inputs correctly, for example that the auto-correlation function and other statistics match.

Reply: We agree that we did not provide evidence of our statement on line 435 of

the original manuscript that the simulated precipitation time series captured the main statistics of the observed time series well. Please find below evidence for this statement (Table 1 and Figure 1. The match is not perfect but we judged it close enough, given that the model is only an approximation of the real world (e.g., it assumes that the log-transformed precipitation has a normal distribution, constant a priori mean and variance, and a stationary near-exponential autocorrelation function that results from an AR(1) model formulation).

**Table 1.** Mean and variance of the log-transformed observed precipitation time series at Esch-sur-Sure and Dahl rain gauges and the simulated precipitation time series at Goesdorf (random selection of simulation numbers 1, 750, 1500 and all).

|          | Esch-sur-Sure | Dahl    | Sim 1   | Sim 750 | Sim 1500 | Sims (All) |
|----------|---------------|---------|---------|---------|----------|------------|
| Mean     | -6.6152       | -6.5817 | -6.3888 | -6.3886 | -6.3878  | -6.3874    |
| Variance | 1.4188        | 1.5731  | 1.5636  | 1.5579  | 1.5594   | 1.5582     |

Table 2 shows comparisons of the means and variances for $C_{COD,S}$ and $C_{NH4,S}$ based on 91 measurements in the Haute-Sure catchment and simulations at Goesdorf (note that for $COD_r$ a comparison cannot be made because we had no measurements of $COD_r$ and a model for $COD_r$ was based on expert judgement). The agreement between observed and simulated statistics is again quite close. We could not evaluate the autocorrelation functions of the observed $C_{COD,S}$ and $C_{NH4,S}$ because there were too few observations to be able to compute these (note that the 91 observations were from multiple locations within the catchment, see original manuscript lines 401-403).

'Uncertainty analysis' is an umbrella term. Therefore I would encourage to use always the more specific terms 'uncertainty propagation' and 'sensitivity analysis' (SA) when referring to these concepts. SA is a well established term for calculating the "contributions of input variables to total uncertainty".

**Table 2.** Mean and variance of log-transformed observed $C_{COD,S}$ and $C_{NH4,S}$ in the Haute-Sure catchment and of log-transformed simulated $C_{COD,S}$ and $C_{NH4,S}$ at Goesdorf (random selection of simulation numbers 1, 750, 1500 and all).

|  | Observations | Sim 1 | Sim 750 | Sim 1500 | Sims (All) |
|---|---|---|---|---|---|
| Mean (log($C_{COD,S}$)) | 4.3783 | 4.3752 | 4.3737 | 4.4106 | 4.3780 |
| Variance (log($C_{COD,S}$)) | 0.5637 | 0.5261 | 0.5257 | 0.5394 | 0.5640 |
| Mean (log($C_{NH4,S}$)) | 1.4733 | 1.4656 | 1.4639 | 1.4865 | 1.4730 |
| Variance (log($C_{NH4,S}$)) | 0.1679 | 0.1704 | 0.1684 | 0.1615 | 0.1681 |

Reply: We agree that uncertainty analysis is an umbrella term and will make sure that we use it only in that way, while we use the terms 'uncertainty propagation' and 'stochastic sensitivity analysis' when we refer to a specific method. We prefer to use 'stochastic sensitivity analysis' instead of just 'sensitivity analysis' because in fact there is a lot of confusion about this term in the literature. Many deterministic modellers interpret sensitivity analysis as an approach that analyses how the model responds to (small) changes in its inputs and parameters, irrespective of how uncertain the inputs or parameters are. In fact that is also how we interpreted 'sensitivity' in Sections 2.4 and 3.1 (when we did the quick-scan). We quote from Smith and Smith (2007, page 71): "Whereas a sensitivity analysis defines how the model responds to changes in its components, an uncertainty analysis determines how much uncertainty is introduced into the model by each component of the model".

We will check the manuscript carefully to make the necessary changes, also in the title (replace 'uncertainty propagation' by 'uncertainty analysis') and some section headings (i.e. Sections 2.6 and 3.3 of the original manuscript).

While hinted a several places, I think it would be beneficial to distinguish the kind of uncertainty that one tries to model more explicitly. Some inputs are intrinsically stochastic (e.g. precipitation), while others the uncertainty expresses our lack of knowledge about

a parameter value.

Reply: In this paper uncertainty is always an expression of limited knowledge about a model input or model parameter. We are well aware that in the literature different types of uncertainty are distinguished (such as epistemic and stochastic, see Refsgaard et al. (2007, Section 3.3)), but we doubt that such distinction is useful and can in fact be made. For example, precipitation is the result of a physical-deterministic process and hence not intrinsically stochastic, no matter that it is practically impossible to know it without error everywhere, all the time. In fact, it is hard to imagine that there are truly stochastic processes in nature (note the fundamental difference with chaotic processes, which do exist but are deterministic). Some argue that there is stochastic uncertainty at the quantum-mechanistic level, but even there, there are doubts (we only need to quote Einstein who said "God does not play dice with the universe"). Our paper intentionally does not want to go this philosophical road, and that is why we refrained from distinguishing different kinds of uncertainty.

* Specific points

** Abstract

L4: the paper does not introduce a framework (which would be a theory about how to deal with uncertainties).

Reply: We acknowledge that the main contribution is the application and we will remove all references to a 'new framework' from the paper. Please see also our reply to your first general comment.

** Introduction

L54: "determinism" is the absence of uncertainty, hence it cannot "represent uncertainty"

Reply: Here we merely cited the literature. But we see your point and have rephrased the sentence to: "Five approaches to represent the presence or absence of uncertainty and how it is represented in the context of urban water systems are often distinguished...".

** Material and methods

L115, eq2: Maybe I missed something, but is there no delay? The rain is transformed immediately to runoff?

That is correct. Indeed, we had not explained in Section 2.1 that a delay is included in EmiStatR (see also Torres-Matallana et al., 2018, Section 2.2.3) and will correct this in the revision. We will replace the text in lines 115-120 with:

The contribution of rainwater to the combined sewage volume, $Q_r$ [m$^3$·s$^{-1}$], is derived from precipitation as follows:

$$Q_{r_t} = \frac{1}{6} \cdot P_{t-t_{fS}} \cdot [C_{imp} \cdot A_{imp} + C_{per} \cdot (A_{total} - A_{imp})] \qquad (1)$$

where $1/6$ is a factor for units conversion, $P_{t-t_{fS}}$ precipitation at time $t - t_{fS}$ [mm· min$^{-1}$]; $t_{fS}$ is a delay in time response related to flow time in the sewer system; $A_{imp}$ is the impervious area of the catchment [ha]; $A_{total}$ is the total area of the catchment [ha]; $C_{imp}$ is the run-off coefficient for impervious areas [-]; and $C_{per}$ is the run-off coefficient for pervious areas [-].

L160: "Some variable" - which one?

Reply: We assume that you refer to line 163: "Some of the variables were calibrated based on observations...". These variables are water consumption ($qs$), infiltration flow ($q_f$), time flow ($t_{fS}$), run-off coefficient for impervious area ($C_{imp}$), run-off coefficient for pervious area ($C_{per}$), orifice coefficient of discharge ($C_d$) and initial water level ($lev_{ini}$). We will include the variable names into the text to be clearer which variables have been calibrated.

Table 1: I would remove all irrelevant "inputs", such as ID, name of structure, ...

Reply: We will delete ID of the structure, Name of the structure, Name of the municipality, Name of the catchment and Number of the catchment from Table 1. We will also change the table caption to: "Most important general, CSO input and output variables of EmiStatR, with base values for the general input variables.".

Table 2: What are "flow time structure" and "curve level"? They do not show up in the model equations

Reply: Thank you for spotting this mistake. Flow time structure ($t_{fS}$) should be included in Eq. 2, this is the delay in time response related to the flow time in the sewer system. See also our reply to your comment above on missing the delay from rain to runoff. The curve level - volume ($lev2vol$) refers to a characteristic of a CSOT and translates the level of the water in the CSOT tank to the CSOT volume. To save space we did not include it in our paper (as mentioned in lines 101-102 of the original submission full details are provided in Torres-Matallana et al., 2018). To avoid confusion we will remove this parameter from Table 2.

L165: This is an example where "uncertainty propagation" should be used instead of

"uncertainty analysis"

Reply: We could not find the term "uncertainty analysis" in line 165. Perhaps the reviewer refers to line 176? We will replace "uncertainty analysis" by "uncertainty propagation analysis" in line 176.

L165ff: Why was the selection of inputs needed at all for the uncertainty propagation? The computational efforts do not change with the number of inputs (they do for the SA). Also there is not needed that the model is sensitive to an input. If it is not, than we will just get smaller uncertainties.

Reply: We agree that from a computational point of view there is no need to reduce the number of inputs. However, each uncertain input needs to be modelled by means of a (complex) probability distribution, and as we indicated in lines 176-177 of the original manuscript this is the most difficult and time-consuming step. From this perspective, it definitely pays off to reduce the number of uncertain inputs and focus on the most important ones.

L179: the equation is for the cdf, not the pdf. Also emphasize on 'marginal' implies that you want to model a joint distribution.

Reply: We had defined 'pdf' in line 175 as a 'probability distribution function'. This can refer to a cumulative probability distribution, a probability mass function (for discretely and categorical variables) and a probability density function (see statistical; text books or https://en.wikipedia.org/wiki/Probability_distribution_function). Eq. 6 is a cumulative probability distribution function, hence in the notation that we introduced in line 175 it is a cumulative pdf.

L188ff: how did you determine the order of the AR process (seems always to be 1)?

[Figure]

Reply: We chose the order to be 1 in all cases to keep the model as simple as possible while still being able to handle autocorrelation.

L199: Not the uncertainties are correlated, but the values themselves

Reply: Thank you for this interesting perspective. However, we think that uncertainties can also be correlated. For instance, suppose we estimated $C_{COD,S}$ and $C_{NH4,S}$ using some model but are uncertain about their true value because of estimation errors. Then it is likely that if we overestimated $C_{COD,S}$ we will also have overestimated $C_{NH4,S}$, because the model may have missed an emission event. Similarly, underestimation of both $C_{COD,S}$ and $C_{NH4,S}$ will also tend to occur simultaneously. Thus, if we wish to model our uncertainty (i.e., the estimation errors) by random variables (one for $C_{COD,S}$ and one for $C_{NH4,S}$), then we would include a positive correlation between the two random variables.

L211: Precipitation lot of zeros. It is not clear how the presented model can describe the dry periods.

Reply: As mentioned in lines 229-231, we applied a Kernel smoothing to the precipitation time series prior to modelling. This removed many of the zeroes. In addition, as explained in line 233, we only used precipitation data for calibration of the AR(1) model if these were above the threshold of 0.01 mm.

L215-255: Remove the description as the model is already described in Torres-Matallana et al. (2017)

Reply: We agree that we provided too much detail. We have moved the text from line 223 onward to the Supporting Information. We prefer to keep the text in lines 215-222 in the main article to explain the reader what we did. We will rephrase this text to:

"Torres-Matallana et al. (2017) present a model to simulate precipitation inside a target catchment given a known precipitation time series in a nearby location outside the catchment, while accounting for the uncertainty that is introduced due to spatial variation in precipitation. The method used for input precipitation uncertainty characterisation is essentially the same as the application of a Kalman filter/smoother (Kalman, 1960; Webster and Heuvelink, 2006). Calibration of the model requires precipitation time series at two locations near the catchment of interest. Once the model is calibrated, it is used to simulate precipitation inside the target catchment from a single precipitation time series nearby the catchment."

L257-260: Remove or move to introduction.

Reply: We agree and will remove these lines.

L304: This is sensitive analysis and does not belong under the section "uncertainty propagation" Reply: We will change the title of Section 2.6 to "Uncertainty analysis".

L317: How did you aggregate for the whole year? Is it a problem that the individual indices are not independent due to the auto-correlation?

Reply: We aggregated to a yearly value by taking the arithmetic mean of all 10 minute indices within the year. We will explain this in the revision. We agree that there likely is temporal structure in the 10 minute indices but that has no effect on the usefulness of computing and interpreting a yearly aggregate. It still is an overall annual measure of the stochastic sensitivity.

L340: "may be computed" - did you do so?

Reply: Thank you for noting our sloppy formulation. We will change this sentence to: "We used the German guideline ATV-A 128 (1992), which computes the throttle discharge at CSOs, $Q_{t,CSO}[l \cdot s^{-1}]$ as:"

** Results

L397: The wording is a bit confusing here: "evaluation of model output sensitivities" sounds like a SA, but you are referring to the "manual" analysis of the model.

Reply: As explained above the term 'sensitivity analysis' has multiple meanings. To many it simply refers to the sensitivity of the model output to changes in the model inputs. This is also the interpretation we used here. To avoid confusion we will rephrase the sentence to "After ranking all inputs on level of uncertainty and model sensitivity, we selected..."

L415: was the COD modeled independently of the precipitation?

You are right, we did not include cross-correlation between $COD_r$ and precipitation, as mentioned in lines 637-638. We agree that it would be more realistic to include a correlation and had mentioned this under 'possible improvements' in lines 637-641.

Figure3: Please mention in the caption which density you used of the uncertainty propagation. Also, maybe move figure to supporting information (SI).

Reply: Thank you for this comment and suggestion. We will move this figure to the supporting information and will extend the caption with: "Note that the blue densities were used in the uncertainty propagation".

L435: Please show some evidence for that.

Reply: Please see our response to your general comment. We are happy to provide the evidence in the Supporting Information.

L444-465: I propose to move this and figure 4 to the SI

We will move Section 3.3.1 to the Supporting Information and refer to it in the main text.

** Discussion

L560: "we start with the accuracy..." - this topic comes second.

Reply: Thank you for noting this mistake, which was also pointed out by Referee 1. We will correct this in the revised manuscript, by changing the text to: "In the following discussion, we start with the uncertainty and water quality impact of the model outputs to the environment, in relation to the uncertainty analysis. Next, we discuss the accuracy of Monte Carlo analysis, followed by a discussion of other sources of uncertainty. Finally, we highlight some limitations and possible solutions of the approach used in this work."

L637: This is an important suggestions. I'm a bit confused why you did not considered the correlation if the model is apparently able to do so ("We used the latest version of EmiStat R (version 1.2.2.0), which considers this kind of patterns.")

Reply: This seems to be a misunderstanding. EmiStatR version 1.2.2.0 can account for $C_{COD,S}$ and $C_{NH4,S}$ to be correlated with the German ATV-A 134 curve, i.e. daily consumption. However, it cannot account for a correlation between $COD_r$ and precipitation.

**References**

Kalman, R. E.: A new approach to linear filtering and prediction problems, Transactions of the American Society of Mechanical Engineers: Journal of Basic Engineering, 82D, 35–45, 1960.

Refsgaard, J. C., van der Sluijs, J. P., Højberg, A. L., and Vanrolleghem, P. A.: Uncertainty in the environmental modelling process - A framework and guidance, Environmental Modelling and Software, 22, 1543–1556, https://doi.org/10.1016/j.envsoft.2007.02.004, 2007.

Smith, J. and Smith, P. Environmental Modelling: An Introduction. ISBN: 9780199272068. Oxford University Press. 2007.

Torres-Matallana, J. A., Leopold, U., and Heuvelink, G. B. M.: Multivariate autoregressive modelling and conditional simulation of precipitation time series for urban water models, European Water, 57, 299–306, 2017.

Torres-Matallana, J. A., Klepiszewski, K., Leopold, U., and Heuvelink, G.: EmiStatR: a simplified and scalable urban water quality model for simulation of combined sewer overflows, Water, 10(6), 1–24, https://doi.org/10.3390/w10060782, 2018.

Webster, R. and Heuvelink, G. B. M.: The Kalman filter for the pedologist's tool kit, European Journal of Soil Science, 57, 758 773, https://doi.org/10.1111/j.1365-2389.2006.00879.x, 2006.
* * *
[Figure]

**Fig. 1.** Autocorrelation function of the log-transformed observed precipitation time series at Esch-sur-Sure and Dahl rain gauges and simulated precipitation at Goesdorf catchment.

---

## Author Response (AR1)

**Reply to interactive comment from Referee 1 on "Multivariate autoregressive modelling and conditional simulation for temporal uncertainty propagation in urban water systems" by Jairo Arturo Torres-Matallana et al.**

Jairo Arturo Torres-Matallana, Ulrich Leopold,
Gerard B.M. Heuvelink

The manuscript "Multivariate autoregressive modelling and conditional simulation for temporal uncertainty propagation in urban water systems" aims to select and characterize the main sources of input uncertainty in urban water systems and quantifying the contributions of each uncertainty source to model output uncertainty over time. It provides a good and well-structured example of an uncertainty analysis for a quite simplified model. Especially the results on CSO water quantity and quality are interesting and useful for further studies. In general, I think the manuscript is a valuable addition to the field with some minor discussion points I would like to bring up:

Reply: Thank you for your kind words and valuable comments that helped us to improve the manuscript. We considered each comment and please find our replies below after each comment from Referee 1.

[1] Page 2 – Line 40: Is the minimization of CSO volume alone a goal in itself? There is the question if many events with a bad water quality (e.g. first flush) are better than fewer events with higher volume and better water quality? That may be a point for elaboration.

Reply: Good point. Minimization of CSO volume is not the only goal. In the revision, we have changed the sentence to: (Revision, Page 2 L38-39) "To reduce pollution in receiving waters it is important to minimise CSO load and concentration".

[2] Page 12 – Line 261: As your model is quite simple and requires "little" computational time the chosen method is feasible, but that is not the case in most of those integrated studies. Is that not a limitation worth mentioning and discussing in 4.4? How could the approach look like in a more complex model?

Reply: We agree that this point should be addressed in the Discussion and have made appropriate adaptations in the revised manuscript in Section 4.4 (Revision, Page 30, L638-647):

> **6. Uncertainty analysis with complex models.** In this research we were able to conduct a comprehensive Monte Carlo uncertainty propagation analysis, which required a large number of Monte Carlo runs. This was possible because we used a strongly simplified urban water system model, EmistatR. For more complex models that take much more computing time, application of a Monte Carlo uncertainty propagation analysis is more challenging. However, given sufficient resources it is possible, because each model run can be run independently and hence the analysis is extremely suitable for parallisation and cloud computing. In particular, the use of graphics processing units (GPU) for heavy computation is promising. Some recent examples that demonstrate the potential of GPU for this purpose are Eranen et al. (2014), Sten et al. (2016) and Sandric et al. (2019). Sriwastava et al. (2018) applied uncertainty propagation to a complex hydrodynamic model, by selecting a small subset of dominant input/model parameters that explain most of the model output variance.

The methodology used in our study may be replicated for a model of higher complexity because of the scalable approach that was followed. The main limitation of application to a higher model complexity case is not the method implementation itself but the hardware setup that is required to make the uncertainty propagation feasible. It is necessary to speed up the computations of a single model run, which is not always an easy task.

[3] Page 19 – Table 4: I quite like this very accessible and clear table for the decision- making of which input variables you select. Still, I think that the variables that are awarded $++$ and $+$ for uncertainty and sensitivity respectively must be discussed more. Especially I think that on the infiltration, NH4 in Rainwater, and C pervious where I don't necessarily agree with omitting them, at least not on the argues in the text of 3.1. On the other hand, I am surprised on the uncertainty of the total area. So, the distinction where the authors draw the deciding line in what to include into their analysis must be clearer. It could be maybe better explained by using graphical panels (e.g. in QUICS (Tscheikner-Gratl et al., 2017)) for illustrating that decision.

Reply: We agree that the decision between $++$ and $+$ for uncertainty and sensitivity needs more justification and have made appropriate adaptations in the revised manuscript for inflow of infiltration water, NH4 in rainwater, and C pervious:

(Revision, Page 14, L326-328) "To better support our decisions we also include a graphical assessment of the degree of uncertainty and sensitivity of each

input, as in Tscheikner-Gratl et al. (2017), see Figure S1 in the Supplementary Material.

Adaptation Section 3.1.2 (Revision, Page 15, L341-343): Regarding the inflow of infiltration water (4), "Although this is a very uncertain input, the quick-scan analysis showed that model output sensitivity is not very high as is indicated in Table 3. For this reason we did not include this variable in the uncertainty propagation analysis."

Adaptation Section 3.1.3 (Revision, Page 15, L353-354): Regarding $NH_4$ in rainwater (9), "Although model output is very sensitive to this model input variable, model input uncertainty is not very high as is indicated in Table 3. For this reason it was not included in the uncertainty propagation analysis."

Adaptation Section 3.1.4 (Revision, Page 15, L359-361): "Although model output is very sensitive to the input variable $C_{per}$ (13), the uncertainty about this variable is not very high, as indicated in Table 3. The reason behind this is that $C_{per}$ can be derived fairly accurately from GIS products, such as land use and soil type maps. Therefore, we did not include this variable in the uncertainty propagation analysis."

[4] Page 29 – Line 560: You don't start with the accuracy of Monte Carlo Analysis (which is then 4.2) but with Uncertainty and water quality impact (4.1).

Reply: Thank you for noting this mistake. We have corrected this in the revised manuscript, by changing the text to: (Revision, Page 25, L515-518) "In the following discussion, we start with the uncertainty and water quality impact of the model outputs to the environment, in relation to the uncertainty analysis. Next, we discuss the accuracy of Monte Carlo analysis, followed by a discussion of other sources of uncertainty. Finally, we highlight some limitations and possible solutions of the approach used in this work."

[5] Page 31 – Line 588: I agree that that is one of the very valuable contribution of this paper. Still I would like to see some comparisons to other attempts on quantity (e.g. Sriwastava et al., 2018) and quality (especially measurements taken at CSOs the measured water quality at the WWTP influent is expected to render a low representativity of the conditions at the CSOs - e.g. Brombach et al.(2005); Diaz-Fierros T et al. (2002))

Reply: Thank you for your kind words and suggestion. We have made appropriate adaptations in the revised manuscript by expanding the text and including comparisons with other quantity and quality studies as follows:

(Revision, Page 27, L544-566)"We also recognise other attempts on quantity (e.g. Sriwastava et al., 2018) and quality, especially measurements taken at CSOs, which demonstrate that the measured water quality at the WWTP influent is expected to render a low representativity of the conditions at the CSOs (e.g. Brombach et al.(2005); Diaz-Fierros T et al. (2002)). We present some comparisons with these studies in the following lines.

Sriwastava et al. (2018) apply uncertainty propagation to a complex hy-

[Figure]

Figure 1: Graphical assessment of the contribution of input uncertainty to model output uncertainty. Numbers near each dot refer to the input variable number as defined in Table 4 of the manuscript. Panel layout after Tscheikner-Gratl et al. (2017).

drodynamic model for quantifying uncertainty in sewer overflow volume. They used MC for uncertainty propagation and Latin hypercube sampling (LHS) as an efficient sampling scheme. Although LHS ensures a full coverage of the sample space and provides a faster convergence than simple random sampling, the LHS application in the case of dynamic model inputs (e.g. precipitation, COD and NH4 inputs) is not trivial and its implementation is more complex than in the case of sampling from static variables (i.e., uncertain constants). In our study, we sampled time series of dynamic inputs using an implementation in stUPscales (Torres-Matallana et al., 2019; Torres-Matallana et al., 2018).

Diaz-Fierros et al. (2002), in a study in the city of Santiago de Compostela (North-West Spain, population about 100,000 inhabitants), where a combined sewer system feeds to a grossly under-sized wastewater treatment plant, reported an event mean concentration (Diaz-Fierros et al. (2002), Table 4) for the output variables $C_{COD,Sv,av}$ and $C_{NH4,Sv,av}$ of 329.1 mg·l$^{-1}$ and 8.7 mg·l$^{-1}$, respectively. These values are larger than those found by Brombach et al. (2005), and more in agreement with our findings, especially for the case of $C_{NH4,Sv,av}$. Diaz-Fierros et al. (2002) reported values of $C_{COD,Sv,av}$ as high as 1073 mg·l$^{-1}$, which agrees with the right-hand tail of the distribution obtained in our study (i.e. a 0.995 quantile of 909.7 mg·l$^{-1}$). Similarly, for the case of $C_{NH4,Sv,av}$, Diaz-Fierros et al. (2002) reported values as high as 32.5 mg·l$^{-1}$, comparable with the 0.995 quantile (29.20 mg·l$^{-1}$) found in our study.

It is worth noting that regarding measurements taken at CSOs, the measured water quality at the WWTP influent is expected to render a low representativity of the conditions at the CSOs as reported by Diaz-Fierros et al. (2002) and Brombach et al. (2005). Thus, when comparing model outputs with independent measurements, one should bear in mind that discrepancies between measured and predicted are not only caused by errors in model inputs, model parameters and model structure, but are also the result of errors in the water quality measurements."

[6] Page 32 – Line 62: The point about linkage is an important one, but I don't see the big input from this paper on the topic. Can you elaborate on this, why is the quantification at sub-module level advisable? Only due to the computational budget limitations?

Reply: We agree that we did not address this aspect in our paper but in the Discussion we did want to point to the possibility of obtaining uncertainties at sub-model level. Some users may be interested at uncertainty levels of sub-modules of the model. For example, sub-module outputs are of particular interest in Bach et al. (2015), Burger et al. (2016) and Rauch et al. (2017).

Perhaps the text on lines 632-636 of the original manuscript was not very clear. We will reformulate it to: (Revision, Page 29, L610-615) "Tscheikner-Gratl et al. (2019) addressed the question as to whether there is an increase in uncertainty by linking integrated models or whether a compensation effect could take place by which overall uncertainty in key water quality parameters decreases. Some further insight into this topic could be obtained by quantifying uncertainties at sub-model level, and analysing whether uncertainty at sub-model level is greater or smaller than at the overall level. With our implementation this is not a difficult task because EmistatR has a stringent modular design in which it is easy to analyse outputs and their uncertainties at sub-model level."

[7] Page 34 - Line 701: Your abstract starts with "Uncertainty is often ignored in urban water systems modelling." I would have therefore expected and would like to read how this can be improved and how studies like yours can provide guidance for the decision makers.

Reply: We believe that we have made a contribution towards making uncertainty propagation analysis in urban water systems modelling more routine. Clearly a single journal publication is not enough but we provide guidance, a simplified model that is very suited for Monte Carlo uncertainty propagation, and we shared the code scripts as well as the datasets to reproduce Figures 3 to 6, so that interested parties could more easily run an uncertainty analysis themselves. Please also note that our study was part of the larger 'QUICS' EU project (https://www.sheffield.ac.uk/quics), which aimed to stimulate the use of uncertainty analysis in integrated catchment modelling, and which involved partners from industry, water management authorities and consultancy firms.

Literature:

Brombach, H., Weiss, G., Fuchs, S., 2005. A new database on urban runoff pollution: comparison of separate and combined sewer systems. Water Sci Technol 51, 119–128. https://doi.org/10.2166/wst.2005.0039

Diaz-Fierros T, F., Puerta, J., Suarez, J., Diaz-Fierros V, F., 2002. Contaminant loads of CSOs at the wastewater treatment plant of a city in NW Spain. Urban Water 4, 291–299. https://doi.org/10.1016/S1462-0758(02)00020-1

Sriwastava, A.K., Tait, S., Schellart, A., Kroll, S., Van Dorpe, M., Van Assel, J., Shucksmith, J., 2018. Quantifying Uncertainty in Simulation of Sewer Overflow Volume. Journal of Environmental Engineering 144, 04018050. https://doi.org/10.1061/(ASCE)EE.1943-7870.0001392

Torres-Matallana, J., Leopold, U., and Heuvelink, G.: stUPscales: an R-package for spatio-temporal Uncertainty Propagation across multiplescales with examples in urban water modelling, Water, 10(7), 1–30, https://doi.org/10.3390/w10070837, 2018.

Torres-Matallana, J., Leopold, U., and Heuvelink, G.: stUPscales: Spatio-Temporal Uncertainty Propagation Across Multiple Scales, https://CRAN.R-project.org/package=stUPscales, r package version 1.0.5.0, 2019.

Tscheikner-Gratl, F., Lepot, M., Moreno-Rodenas, A., Schellart, A., 2017. A Framework for the application of Uncertainty Analysis (Deliverable No. 6.7), QUICS. Zenodo, https://zenodo.org/record/1240926

**Reply to interactive comment from Referee 2 on "Multivariate autoregressive modelling and conditional simulation for temporal uncertainty propagation in urban water systems" by Jairo Arturo Torres-Matallana et al.**

Jairo Arturo Torres-Matallana, Ulrich Leopold,
Gerard B.M. Heuvelink

Reply: We welcome your comments, which were very helpful to improve our manuscript. In the text below we provide our replies (in blue) after each original comment.

* General
This manuscript presents a detailed case study on uncertainty propagation through a water quality model. The authors propose the use of auto-regressive models to describe the dynamic of the input time-series.

For the reader it is currently unclear if the focus is on the method ("this paper introduces an uncertainty analysis framework") or the application case. My suggestion is to focus on the case study, as the methodological contribution is rather limited. In any case, the focus should be set clearer in the introduction. In general, focusing more on the key points by moving some material to the supporting information would help.

Reply: We agree that the added value of the paper is mainly in the application, and we have made this clear in the revision, for example by no longer claiming that we introduce an uncertainty analysis framework but instead recognising that we build on existing methods (Revision, Page 1, L4-5). To emphasise that the application is most important we also changed the title of the manuscript to "Multivariate autoregressive modelling and conditional simulation for temporal uncertainty analysis of an urban water system **in Luxembourg**".

Even though the focus is on the application we do think that our paper also makes a valid contribution on methods. For example, we use both univariate and multivariate autoregressive models to characterise uncertainty in dynamic variables, we use conditional simulation to sample from these models, and we use bootstrap computation to summarise the Monte Carlo outputs. None of

these methods are new but as far as we know they have not been used jointly in uncertainty analysis of urban water systems. We have moved some of the materials to the Supplementary Material, in particular the second part of Section 2.5.4 where we explain the calibration and conditional simulation of the precipitation time series (i.e., lines 223 to 255 in the original manuscript).

The authors use AR(1) processes to model dynamical inputs. However, it needs also to be shown that these models captures the characteristics of the inputs correctly, for example that the auto-correlation function and other statistics match.

Reply: We agree that we did not provide evidence of our statement on line 435 of the original manuscript that the simulated precipitation time series captured the main statistics of the observed time series well. Please find below evidence for this statement (Table 1 and Figure 1). The match is not perfect but we judge it close enough, given that the model is only an approximation of the real world (e.g., it assumes that the log-transformed precipitation has a normal distribution, constant a priori mean and variance, and a stationary near-exponential autocorrelation function that results from an AR(1) model formulation). We made a note in the text (Revision, Page 19, L411-412) and refer to the Supplementary Material, Table S1 and Figure S2.

Table 1: Mean and variance of the log-transformed observed precipitation time series at Esch-sur-Sure and Dahl rain gauges and the simulated precipitation time series at Goesdorf (random selection of simulation numbers 1, 750, 1500 and all).

|          | Esch-sur-Sure | Dahl    | Sim 1   | Sim 750 | Sim 1500 | Sims (All) |
|----------|---------------|---------|---------|---------|----------|------------|
| Mean     | -6.6152       | -6.5817 | -6.3888 | -6.3886 | -6.3878  | -6.3874    |
| Variance | 1.4188        | 1.5731  | 1.5636  | 1.5579  | 1.5594   | 1.5582     |

Table 2 shows comparisons of the means and variances for $C_{COD,S}$ and $C_{NH4,S}$ based on 91 measurements in the Haute-Sure catchment and simulations at Goesdorf (note that for $COD_r$ a comparison cannot be made because we had no measurements of $COD_r$ and a model for $COD_r$ was based on expert judgement). The agreement between observed and simulated statistics is again quite close. We could not evaluate the autocorrelation functions of the observed $C_{COD,S}$ and $C_{NH4,S}$ because there were too few observations to be able to compute these (note that the 91 observations were from multiple locations within the catchment, see original manuscript lines 401-403).

'Uncertainty analysis' is an umbrella term. Therefore I would encourage to use always the more specific terms 'uncertainty propagation' and 'sensitivity analysis' (SA) when referring to these concepts. SA is a well established term for calculating the "contributions of input variables to total uncertainty".

Reply: We agree that uncertainty analysis is an umbrella term and made sure that we used it only in that way, while we use the terms 'uncertainty prop-

[Figure]

Figure 1: Autocorrelation function of the log-transformed observed precipitation time series at Esch-sur-Sûre and Dahl rain gauges and simulated precipitation at Goesdorf catchment.

Table 2: Mean and variance of log-transformed observed $C_{COD,S}$ and $C_{NH4,S}$ in the Haute-Sure catchment and of log-transformed simulated $C_{COD,S}$ and $C_{NH4,S}$ at Goesdorf (random selection of simulation numbers 1, 750, 1500 and all).

|  | Observations | Sim 1 | Sim 750 | Sim 1500 | Sims (All) |
|---|---|---|---|---|---|
| Mean $(\log(C_{COD,S}))$ | 4.3783 | 4.3752 | 4.3737 | 4.4106 | 4.3780 |
| Variance $(\log(C_{COD,S}))$ | 0.5637 | 0.5261 | 0.5257 | 0.5394 | 0.5640 |
| Mean $(\log(C_{NH4,S}))$ | 1.4733 | 1.4656 | 1.4639 | 1.4865 | 1.4730 |
| Variance $(\log(C_{NH4,S}))$ | 0.1679 | 0.1704 | 0.1684 | 0.1615 | 0.1681 |

agation' and 'stochastic sensitivity analysis' when we refer to a specific method. We prefer to use 'stochastic sensitivity analysis' instead of just 'sensitivity analysis' because in fact there is a lot of confusion about this term in the literature. Many deterministic modellers interpret sensitivity analysis as an approach that analyses how the model responds to (small) changes in its inputs and parameters, irrespective of how uncertain the inputs or parameters are. In fact that is also how we interpreted 'sensitivity' in Sections 2.4 and 3.1 (when we did the quick-scan). We quote from Smith and Smith (2007, page 71): "Whereas a sensitivity analysis defines how the model responds to changes in its components, an uncertainty analysis determines how much uncertainty is introduced into the model by each component of the model".

We checked the manuscript carefully to make the necessary changes, also in the title (replaced 'uncertainty propagation' by 'uncertainty analysis') and some section headings (i.e. Sections 2.6 (Revision, Page 11, L225) and 3.3 Revision, Page 19, L417) in the revised manuscript).

While hinted a several places, I think it would be beneficial to distinguish the kind of uncertainty that one tries to model more explicitly. Some inputs are intrinsically stochastic (e.g. precipitation), while others the uncertainty expresses our lack of knowledge about a parameter value.

Reply: In this paper uncertainty is always an expression of limited knowledge about a model input or model parameter. We are well aware that in the literature different types of uncertainty are distinguished (such as epistemic and stochastic, see Refsgaard et al. (2007, Section 3.3)), but we doubt that such distinction is useful and can in fact be made. For example, precipitation is the result of a physical-deterministic process and hence not intrinsically stochastic, no matter that it is practically impossible to know it without error everywhere, all the time. In fact, it is hard to imagine that there are truly stochastic processes in nature (note the fundamental difference with chaotic processes, which do exist but are deterministic). Some argue that there is stochastic uncertainty at the quantum-mechanistic level, but even there, there are doubts (we only need to quote Einstein who said "God does not play dice with the universe"). Our paper intentionally does not go this philosophical road, and that is why we refrained from distinguishing different kinds of uncertainty.

* Specific points

** Abstract

L4: the paper does not introduce a framework (which would be a theory about how to deal with uncertainties).

Reply: We acknowledge that the main contribution is the application and have removed all references to a 'new framework' from the paper. Please see also our reply to your first general comment.

** Introduction

L54: "determinism" is the absence of uncertainty, hence it cannot "represent uncertainty"

Reply: Here we merely cited the literature. But we see your point and have rephrased the sentence to: (Revision, Page 2, L53-54)"Five approaches to represent the presence or absence of uncertainty and how it is represented in the context of urban water systems are often distinguished. . . .".

** Material and methods

L115, eq2: Maybe I missed something, but is there no delay? The rain is transformed immediately to runoff?

That is correct. Indeed, we had not explained in Section 2.1 that a delay is included in EmiStatR (see also Torres-Matallana et al., 2018, Section 2.2.3) and have corrected this in the revision. We replaced the text in lines 116-120 with:

(Revision, Page 4, L116-120) The contribution of rainwater to the combined sewage volume, $Q_r$ [m$^3 \cdot$s$^{-1}$], is derived from precipitation as follows:

$$Q_{r_t} = \frac{1}{6} \cdot P_{t-t_{fS}} \cdot [C_{imp} \cdot A_{imp} + C_{per} \cdot (A_{total} - A_{imp})] \tag{1}$$

where $\frac{1}{6}$ is a factor for units conversion, $P_{t-t_{fS}}$ precipitation at time $t - t_{fS}$ [mm$\cdot$ min$^{-1}$]; $t_{fS}$ is a delay in time response related to flow time in the sewer system; $A_{imp}$ is the impervious area of the catchment [ha]; $A_{total}$ is the total area of the catchment [ha]; $C_{imp}$ is the run-off coefficient for impervious areas [-]; and $C_{per}$ is the run-off coefficient for pervious areas [-].

L160: "Some variable" - which one?

Reply: We assume that you refer to line 163: "Some of the variables were calibrated based on observations...". These variables are water consumption ($qs$), infiltration flow ($q_f$), time flow ($t_{fS}$), run-off coefficient for impervious area ($C_{imp}$), run-off coefficient for pervious area ($C_{per}$), orifice coefficient of discharge ($C_d$) and initial water level ($lev_{ini}$). We included the variable names into the text to be clearer which variables have been calibrated (Revision, Page 7, L165-167).

Table 1: I would remove all irrelevant "inputs", such as ID, name of structure, ...

Reply: (Revision, Page 8, Table 1) We deleted ID of the structure, Name of the structure, Name of the municipality, Name of the catchment and Number of the catchment from Table 1. We also changed the table caption to: "Most important general, CSO input and output variables of EmiStatR, with base values for the general input variables.".

Table 2: What are "flow time structure" and "curve level"? They do not show up in the model equations

Reply: Thank you for spotting this mistake. Flow time structure ($t_{fS}$) should be included in Eq. 2 in the original manuscript, this is the delay in time response related to the flow time in the sewer system. See also our reply to your comment above on missing the delay from rain to runoff. The curve level - volume ($lev2vol$) refers to a characteristic of a CSOT and translates the level of the water in the CSOT tank to the CSOT volume. To save space we did not include it in our paper (as mentioned in lines 101-102 of the original submission full details are provided in Torres-Matallana et al., 2018). To avoid confusion we removed this parameter from Tables 1 (Revision, Page 8) and 2 (Revision, Page 9).

L165: This is an example where "uncertainty propagation" should be used instead of "uncertainty analysis"

Reply: We could not find the term "uncertainty analysis" in line 165. Perhaps the reviewer refers to line 176? We have replaced "uncertainty analysis" by "uncertainty propagation analysis" in line 176 of the original manuscript (Revision: Page 9, L179).

L165ff: Why was the selection of inputs needed at all for the uncertainty propagation? The computational efforts do not change with the number of inputs (they do for the SA). Also there is not needed that the model is sensitive to an input. If it is not, than we will just get smaller uncertainties.

Reply: We agree that from a computational point of view there is no need to reduce the number of inputs. However, each uncertain input needs to be modelled by means of a (complex) probability distribution, and as we indicated in lines 176-177 of the original manuscript this is the most difficult and time-consuming step. From this perspective, it definitely pays off to reduce the number of uncertain inputs and focus on the most important ones.

L179: the equation is for the cdf, not the pdf. Also emphasize on 'marginal' implies that you want to model a joint distribution.

Reply: We had defined 'pdf' in line 175 as a 'probability distribution function'. This can refer to a cumulative probability distribution, a probability mass function (for discretely and categorical variables) and a probability density function (see statistical text books or https://en.wikipedia.org/wiki/Probability_distribution_function). Eq. 6 is a cumulative probability distribution function, hence in the notation that we introduced in line 175 it is a cumulative pdf.

L188ff: how did you determine the order of the AR process (seems always to be 1)?

Reply: We chose the order to be 1 in all cases to keep the model as simple as possible while still being able to handle autocorrelation.

L199: Not the uncertainties are correlated, but the values themselves

Reply: Thank you for this interesting perspective. However, we think that uncertainties can also be correlated. For instance, suppose we estimated $C_{COD,S}$ and $C_{NH4,S}$ using some model but are uncertain about their true value because of estimation errors. Then it is likely that if we overestimated $C_{COD,S}$ we will also have overestimated $C_{NH4,S}$, because the model may have missed an emission event. Similarly, underestimation of both $C_{COD,S}$ and $C_{NH4,S}$ will also tend to occur simultaneously. Thus, if we wish to model our uncertainty (i.e., the estimation errors) by random variables (one for $C_{COD,S}$ and one for $C_{NH4,S}$), then we would include a positive correlation between the two random variables.

L211: Precipitation lot of zeros. It is not clear how the presented model can describe the dry periods.

Reply: As mentioned in lines 229-231 of the original manuscript, we applied a Kernel smoothing to the precipitation time series prior to modelling. This removed many of the zeroes. In addition, as explained in line 233 of the original manuscript, we only used precipitation data for calibration of the AR(1) model if these were above the threshold of 0.01 mm.

L215-255: Remove the description as the model is already described in Torres-Matallana et al. (2017)

Reply: We agree that we provided too much detail. We have moved the text from line 223 onward to the Supplementary Material. We prefer to keep the text in lines 215-222 of the original manuscript in the main article to explain the reader what we did. We have rephrased this text to:

> (Revision, Page 11, L218-224) "Torres-Matallana et al. (2017) present a model to simulate precipitation inside a target catchment given a known precipitation time series in a nearby location outside the catchment, while accounting for the uncertainty that is introduced due to spatial variation in precipitation. The method used for input precipitation uncertainty characterisation is essentially the same as the application of a Kalman filter/smoother (Kalman, 1960; Webster and Heuvelink, 2006). Calibration of the model requires precipitation time series at two locations near the catchment of interest. Once the model is calibrated, it is used to simulate precipitation inside the target catchment from a single precipitation time series nearby the catchment. Details of the calibration and conditional simulation are presented in the Section S3 of the Supplementary Material."

L257-260: Remove or move to introduction.
Reply: We agree and have removed these lines.

L304: This is sensitive analysis and does not belong under the section "uncertainty propagation"

Reply: We changed the title of Section 2.6 to (Revision, Page 11, L225) "Uncertainty analysis".

L317: How did you aggregate for the whole year? Is it a problem that the individual indices are not independent due to the auto-correlation?

Reply: We aggregated to a yearly value by taking the arithmetic mean of all 10 minute indices within the year. We explain this in the revision. We agree that there likely is temporal structure in the 10 minute indices but that has no

effect on the usefulness of computing and interpreting a yearly aggregate. It still is an overall annual measure of the stochastic sensitivity.

L340: "may be computed" - did you do so?
Reply: Thank you for noting our sloppy formulation. We changed this sentence to: (Revision, Page 14, L306) "We used the German guideline ATV-A 128 (1992), which computes the throttle discharge at CSOs, $Q_{t,CSO}[l \cdot s^{-1}]$ as:"

** Results
L397: The wording is a bit confusing here: "evaluation of model output sensitivities" sounds like a SA, but you are referring to the "manual" analysis of the model.
Reply: As explained above the term 'sensitivity analysis' has multiple meanings. To many it simply refers to the sensitivity of the model output to changes in the model inputs. This is also the interpretation we used here. To avoid confusion we rephrased the sentence to (Revision Page 16, L371) "After ranking all inputs on level of uncertainty and model sensitivity, we selected..."

L415: was the COD modeled independently of the precipitation?
You are right, we did not include cross-correlation between $COD_r$ and precipitation, as mentioned in lines 637-638. We agree that it would be more realistic to include a correlation and had mentioned this under 'possible improvements' in lines 637-641 of the original manuscript (Revision Page 29, L616-623).

Figure3: Please mention in the caption which density you used of the uncertainty propagation. Also, maybe move figure to supporting information (SI).
Reply: Thank you for this comment and suggestion. We have moved this figure to the Supplementary Material (Page 6, Figure S3) and extended the caption with: "Note that the blue densities were used in the uncertainty propagation".

L435: Please show some evidence for that.
Reply: Please see our response to your general comment. We provided the evidence in the Supplementary Material (Pages 1 and 3, Section S2).

L444-465: I propose to move this and figure 4 to the SI
We have moved Section 3.3.1 to the Supplementary Material (Page 5 and 7-8, Section S5) and refer to it in the main text (Revision, Page 19, L424-425).

** Discussion
L560: "we start with the accuracy..." - this topic comes second.
Reply: Thank you for noting this mistake, which was also pointed out by Referee 1. We have corrected this in the revised manuscript, by changing the

text to: (Revision, Page 25, L515-518) "In the following discussion, we start with the uncertainty and water quality impact of the model outputs to the environment, in relation to the uncertainty analysis. Next, we discuss the accuracy of Monte Carlo analysis, followed by a discussion of other sources of uncertainty. Finally, we highlight some limitations and possible solutions of the approach used in this work."

L637: This is an important suggestions. I'm a bit confused why you did not considered the correlation if the model is apparently able to do so ("We used the latest version of EmiStat R (version 1.2.2.0), which considers this kind of patterns.")

Reply: This seems to be a misunderstanding. EmiStatR version 1.2.2.0 can account for $C_{COD,S}$ and $C_{NH4,S}$ to be correlated with the German ATV-A 134 curve, i.e. daily consumption. However, it cannot account for a correlation between $COD_r$ and precipitation.

** References

[revised manuscript text omitted]

---

## Author Response (AR2)

**Reply to technical corrections requested by Referee 2 on "Multivariate autoregressive modelling and conditional simulation for temporal uncertainty analysis of an urban water system in Luxembourg" by Jairo Arturo Torres-Matallana et al.**

Jairo Arturo Torres-Matallana, Ulrich Leopold,
Gerard B.M. Heuvelink

Dear Dr. Nadav Peleg,

We are very pleased with the conditional acceptance of our manuscript. We moved Tables 1 and 2 to the Supplementary Information as suggested. We also addressed the two remaining small technical comments raised by Referee 2. Please see our replies (in blue) after each original comment.

For clarification we also added Appendix A at the end of the paper, which lists all EmiStatR inputs and outputs. We thought it useful to include this list because it provides the reader with a clear overview and reference of most variables used, which became less accessible after moving Tables 1 and 2 to the Supplementary Information. We noticed it is not uncommon to include such appendix in HESS publications and hope that you agree that it has added value.

Referee 2: Thanks for the revisions. Most unclear points are now clarified. The only points I'm still unsure are (line numbers refer to the one used in the authors reply):

L115: The new equation (1) means that the runoff from EmiStatR has the same shape as the precipitation (with an offset). Is there no "smoothing" or convolution involved at all? Even the simplest bucket model has this feature.

Reply: the Referee is right that we do not include a convolution when transforming rainfall to runoff, but only a delay. In a new release of EmistatR we will incorporate convolution. We have now stated this in the revised manuscript in page 28, lines 649-651.

L211: If I understand your reply you only model wet periods. If this is correct, it would be helpful to state it explicitly.

Reply: Perhaps this point is not totally clear. We simulate the entire time period in EmiStatR and hence model both dry and wet periods. In our reply to your comment L211 we explained that precipitation data were smoothed using a Kernel filter prior to modelling and that for calibration of the AR(1) precipitation model we only used periods with precipitation of 0.01 mm or higher. But we simulated the precipitation time series ensemble for uncertainty propagation for the entire year, which includes wet and dry periods.

[revised manuscript text omitted]

Legend

ASTA stations  Network

🔵 Rain gauge    Conduit

- - - Pump

△ CSO structure

△ WWTP

Google
Satellite

Location of Combined Sewer Overflow structures

1. Goesdorf (GOE)

2. Kaundorf (KAU)

3. Nocher-Route (NOR)

Belgium   Germany
Luxembourg
France

**Figure 2.** The three Haute-Sûre sub-catchments and locations of  CSO structures considered in this study. The background map is provided by © Google Maps.

mode (see Section 3.1). Some of the variables were calibrated based on observations in the  CSOC to simulate water level and concentrations and loads of pollutants spilled in the CSO to the stream, river or lake. These variables are water consumption ($qs$), infiltration flow ($q_f$), time flow ($t_{fS}$), run-off coefficient for impervious area ($C_{imp}$), run-off coefficient for pervious area ($C_{per}$), orifice coefficient of discharge ($C_d$) and initial water level ($lev_{ini}$).

~~General input Units Base value CSO input Units 1. Wastewater1. Catchment dataWater consumption, $qs$ [l(PE$^a\cdot$d)$^{-1}$] 152 Total area, $A_{total}$ haPollution COD$^b$, $C_{COD,S}$ [g(PE$\cdot$d)$^{-1}$] 104.2 Impervious area, $A_{imp}$ haPollution NH$_4^c$, $C_{NH4,S}$ [g(PE$\cdot$d)$^{-1}$] 4.7 Run-off coeff.$^d$ for impervious area, $C_{imp}$ -2. Infiltration waterRun-off coeff. for pervious area, $C_{per}$ -Inflow, $q_f$ l(s$\cdot$ha)$^{-1}$0.116 Flow time structure, $t_{fS}$ time stepPollution COD, $COD_f$ [g(PE$\cdot$d)$^{-1}$] 0 Population equivalents, $pe$ PEPollution NH$_4$, $NH4_f$ [g(PE$\cdot$d)$^{-1}$] 0 2. CSO structure data3. RainwaterVolume, $V$ m$^3$Rain time series, $P$ mmInitial water level, $Lev_{ini}$ mPollution COD, $COD_r$ [mg$\cdot$l$^{-1}$] 71.0 Maximum throttled outflow, $Q_{d,max}$ l$\cdot$s$^{-1}$Pollution NH$_4$, $NH4_r$ [mg$\cdot$l$^{-1}$]~~

~~2.0 Orifice diameter, $D_d$ mOrifice coefficient of discharge, $C_d$ -Output variables 1. QuantityVolume in the CSO chamber, $V_{Chamber}$ m³Overflow spill volume, $V_{Sv}$ m³Overflow spill flow, $Q_{Sv}$ [l·s⁻¹] 2. QualitySpill COD load, $B_{COD,Sv}$ gAverage spill COD conc.ᵉ, $C_{COD,Sv,av}$ [mg·l⁻¹] 99.9th perc.ᶠ spill COD conc., $C_{COD,Sv,99.9}$ [mg·l⁻¹] Maximum overflow COD conc., $C_{COD,Sv,max}$ [mg·l⁻¹] Spill NH₄ load, $B_{NH4,Sv}$ gAverage spill NH₄ conc., $C_{NH4,Sv,av}$ [mg·l⁻¹] 99.9th perc. spill NH₄ conc., $C_{NH4,Sv,99.9}$ [mg·l⁻¹] Maximum spill NH₄ conc., $C_{NH4,Sv,max}$ [mg·l⁻¹]~~

~~The CSO structure input data for the EmiStatRmodel, after calibration. Structures 2 and 3, only $C_d$ was calibrated. CSO input 1. IdentificationID of the structure 1 2 3 Name of the structure FBH Goesdorf FBN Kaundorf FBH Nocher-Route 2. Catchment dataName of the municipality Goesdorf Kaundorf Nocher-Route Name of the catchment Haute-Sûre Haute-Sûre Haute-Sûre Number of the catchment 1 1 1 Land Useᵃ R/I R/I R/I Total area, $A_{total}$ ha30.0 22 18.6 Impervious area, $A_{imp}$ ha5.0 11.0 4.3 Run-off coefficient for impervious area, $C_{imp}$ -0.28 0.3 0.3 Run-off coefficient for pervious area, $C_{per}$ -0.07 0.10 0.10 Flow time structure, $t_{fS}$ time step1 2 2 Population equivalents, $pe$ PE611 358 326 3. CSO structure dataVolume, V m³190 180 157 Initial water level, $Lev_{ini}$ m0.57 1.8 1.8 Maximum throttled outflow, $Q_{d,max}$ 1·s⁻¹5.0 9 4 Orifice diameter, $D_d$, m0.15 0.20 – Orifice coefficient of discharge, $C_d$ -0.67 0.67 0.67~~

[revised manuscript text omitted]